# Optochemical control of slow-wave sleep in the nucleus accumbens of male mice by a photoactivatable allosteric modulator of adenosine A$_{2A}$ receptors

Koustav Roy [1,11], Xuzhao Zhou [1,2,11], Rintaro Otani [1,11], Ping-Chuan Yuan[3,4,11], Shuji Ioka [1], Kaspar E. Vogt [1], Tamae Kondo[1], Nouran H. T. Farag [1], Haruto Ijiri[1,5], Zhaofa Wu [6], Youhei Chitose [7], Mao Amezawa [1], David S. Uygun [8], Yoan Cherasse [1], Hiroshi Nagase [1], Yulong Li[9], Masashi Yanagisawa [1], Manabu Abe [7], Radhika Basheer [8], Yi-Qun Wang [3] ✉, Tsuyoshi Saitoh [1,10] ✉ & Michael Lazarus [1,10] ✉

Optochemistry, an emerging pharmacologic approach in which light is used to selectively activate or deactivate molecules, has the potential to alleviate symptoms, cure diseases, and improve quality of life while preventing uncontrolled drug effects. The development of in-vivo applications for opto-chemistry to render brain cells photoresponsive without relying on genetic engineering has been progressing slowly. The nucleus accumbens (NAc) is a region for the regulation of slow-wave sleep (SWS) through the integration of motivational stimuli. Adenosine emerges as a promising candidate molecule for activating indirect pathway neurons of the NAc expressing adenosine A$_{2A}$ receptors (A$_{2A}$Rs) to induce SWS. Here, we developed a brain-permeable positive allosteric modulator of A$_{2A}$Rs (A$_{2A}$R PAM) that can be rapidly photo-activated with visible light ($\lambda > 400$ nm) and used it optoallosterically to induce SWS in the NAc of freely behaving male mice by increasing the activity of extracellular adenosine derived from astrocytic and neuronal activity.

The 21st century is commonly referred to as the age of light. Opto-chemistry is a new approach to pharmacology in which photosensitive molecules are activated or deactivated with light to evoke physiologic responses with spatial and temporal control[1–4]. The use of light-sensitive photopharmacotherapy may offer the possibility to cure diseases and alleviate symptoms while preventing uncontrolled drug activity as the drug is only active at the times and sites where it pro-duces its therapeutic effect. Although chemical photo-uncaging has been used extensively for in-vitro applications such as patch-clamp electrophysiology, the application of optochemistry in the mammalian brain is hampered by the inaccessibility of the brain to light irradiation,

phototoxic damage to brain tissue by ultraviolet (UV) light typically used to activate photolabile-protecting groups (PPGs), and inadequate delivery of photocaged compounds across the blood-brain barrier (BBB). The in-vivo application of compounds that render neurons or glia photoresponsive without the need for genetic engineering, therefore, has been slow to develop.

The therapeutic potential of targeting adenosine A$_{2A}$ receptors (A$_{2A}$Rs) is significant due to their broad expression in the body and central nervous system (CNS). Numerous A$_{2A}$R agonist and antagonist molecules are reported[5–7], many of which are currently under clinical trials or have already been approved for treatment. The development

A full list of affiliations appears at the end of the paper. ✉e-mail: yiqunwang@fudan.edu.cn; saito.tsuyoshi.gf@u.tsukuba.ac.jp; lazarus.michael.ka@u.tsukuba.ac.jp

of adenosine analogs as $A_{2A}R$ agonists for treating CNS disorders is impeded, however, by poor to non-existent drug transport across the BBB and peripheral side effects such as cardiovascular impairment[8]. We previously developed a positive allosteric modulator (PAM) of $A_{2A}Rs$ ($A_{2A}RPAM-1$) that induces slow-wave sleep (SWS) without affecting cardiovascular function, unlike classic $A_{2A}R$ agonists[9,10]. The nucleus accumbens (NAc) is a novel region for SWS regulation that integrates motivational stimuli[11]. Adenosine is a possible candidate molecule for activating NAc indirect pathway neurons expressing $A_{2A}Rs$ to trigger SWS, as caffeine produces its arousal effect in the NAc by blocking $A_{2A}Rs$[12,13], but direct proof is lacking.

Here we report a brain-permeable and photoactivatable analog of $A_{2A}RPAM-1$ (A400-$A_{2A}RPAM-1$ or opto$A_{2A}RPAM-2$), which allows for selective optoallosteric activation of NAc $A_{2A}R$-expressing indirect pathway neurons that induce SWS in freely behaving mice. Our findings provide proof of concept for the efficacy of coumarin-based optochemistry in eliciting brain responses and open the door to the therapeutic use of these chemicals to treat neurologic disorders.

## Results

### Astrocytic and neuronal activity in the NAc increases extracellular adenosine levels

To investigate whether adenosine and its downstream targets are involved in the control of sleep in the NAc, we first microinjected adenosine, the $A_{2A}R$ agonist CGS 21680, the $A_{2A}R$ antagonist ZM 241385 and the adenosine $A_1$ receptor ($A_1R$) agonist $N^6$-cyclopentyladenosine (CPA) into the NAc of freely behaving mice (Fig. 1a and Supplementary Fig. 1a). We implanted cannulas bilaterally into the NAc of wild type (WT) or $A_{2A}R$ knockout (KO) mice and analyzed electroencephalogram (EEG) and electromyogram (EMG) recordings made after focal NAc injections (2 μL/side) of vehicle, 3.5 mM adenosine, 500 μM CGS 21680, and 50 μM CPA at 19:00 to observe sleep induction when the mice were mostly awake or 6 mM ZM 241385 at 8:30 to observe wake induction when the mice were mostly asleep (Fig. 1b). We confirmed the location of the drug infusions in the NAc by injecting the same 2 μL volume of a 4% solution of fluorescein (Fig. 1c). Compared with vehicle injection, injection of adenosine or CGS 21680 significantly increased SWS for 5 h after microinjection, whereas SWS was not induced after CPA microinjection (Fig. 1b, unpaired 2-tailed Student's t-test and Supplementary Fig. 1b, 2-way repeated measures ANOVA-Bonferroni's multiple comparisons, $F_{(33,198)} = 3.403$, $P < 0.0001$). To assess whether EEG activity was altered by adenosine administration, we compared the normalized EEG power spectrum of SWS in mice treated with vehicle or adenosine (Supplementary Fig. 1c, 2-way repeated measures ANOVA-Bonferroni's multiple comparisons). EEG activity in the frequency range of 0.5–25 Hz during SWS was indistinguishable between adenosine-induced and natural (vehicle injection) SWS. These data suggest that adenosine induced physiologic sleep rather than abnormal sleep. In contrast, when we made focal NAc injections with highly concentrated CPA (500 μM), the SWS was reduced, although the reduction was not significant compared to the vehicle injection (Supplementary Fig. 1d, unpaired Mann-Whitney test). Moreover, focal NAc injections of adenosine did not alter SWS in $A_{2A}R$ KO mice and the $A_{2A}R$ antagonist ZM 241385 reduced SWS compared to vehicle injection (Fig. 1b, unpaired Mann-Whitney test and Supplementary Fig. 1e, f). These results suggest that $A_{2A}Rs$, but not $A_1Rs$, regulate sleep in the NAc.

To examine the activity of NAc astrocytes in sleep-wake regulation, we chemogenetically activated glial fibrillary acidic protein (GFAP)-positive cells in the NAc core of mice by stereotaxic microinjection of adeno-associated virus (AAV) carrying hM3Dq DREADD, which stands for "designer receptors exclusively activated by designer drugs"[14], (AAV-GFAP-hM3DqDREADD; Fig. 1d). First, we evaluated the specificity of hM3Dq DREADD expression in AAV-injected WT mice by immunohistochemical investigation of the expression of the neuronal

marker NeuN, the astrocytic marker GFAP and the AAV-reporter protein mCherry (Fig. 1e). We detected mCherry expression only in GFAP but not NeuN positive cells, suggesting that all infected cells were astrocytes. Next, we stereotaxically microinjected AAV-GFAP-hM3DqDREADD bilaterally into the NAc core of WT and $A_{2A}R$ KO mice (Fig. 1f). Three weeks after the AAV injections, EEG and EMG recordings of the mice were analyzed to assess the sleep/wake behavior of the animals after the intraperitoneal injection of clozapine-N-oxide (CNO). We found that SWS was significantly increased for 6 h in a dose-dependent manner after injecting different doses of CNO compared with the saline injections (Fig. 1g, i; Fig. 1g, $F_{(33,220)} = 1.243$, $P = 0.1817$, 2-way repeated measures ANOVA-Tukey test; Fig. 1i, $F_{(5,30)} = 26.55$, $P < 0.0001$, one way ANOVA-Bonferroni's multiple comparisons). In contrast, treatment of AAV-injected $A_{2A}R$ KO mice with the maximal effective dose of CNO (0.3 mg kg$^{-1}$) failed to induce sleep compared to the vehicle control (Fig. 1h, i), suggesting that SWS induction by astrocytic activity is dependent on adenosine.

We, therefore, investigated extracellular adenosine levels after chemogenetic stimulation of astrocytes using in-vivo microdialysis in freely behaving mice (Fig. 1d, j, k). The dialysates were collected between 19:00 and 21:00 when there was significantly increased SWS and analyzed by ultra-high-performance liquid chromatography (UPLC). Dialysates after saline or CNO injections were collected by inserting the same probe in contralateral sites of the mouse brain. The adenosine concentration was determined by comparison with adenosine standards and normalized between the samples taken after the vehicle and CNO treatments due to the varying recovery rates of the microdialysis probes. The position of the microdialysis probe was confirmed by immunohistochemical analysis of the expression of the AAV reporter protein mCherry (Fig. 1j). Adenosine levels were significantly increased in WT and $A_{2A}R$ KO mice injected with AAV-GFAP-hM3DqDREADD and treated with 0.3 mg kg$^{-1}$ CNO compared to WT mice with astrocytic mCherry expression (Fig. 1k, $F_{(3,12)} = 4.721$, $P = 0.0212$, one way ANOVA-Bonferroni's multiple comparisons). We also stereotaxically injected bilaterally hM3Dq DREADD-expressing AAV containing the neuron-specific human synapsin promoter into the NAc of WT mice and found that adenosine levels were significantly increased after treatment with 0.3 mg kg$^{-1}$ CNO (Fig. 1k).

We also used a highly sensitive and selective G protein-coupled receptor (GPCR)-activation-based (GRAB) adenosine sensor GRAB$_{Ado1.0}$ (Ado1.0)[15,16] to monitor changes in extracellular adenosine levels during spontaneous sleep/wake cycles and in response to chemogenetic activation of astrocytes or neurons (Fig. 2). Following the injection of an AAV expressing Ado1.0 or its non-binding mutant GRAB$_{Ado1.0mut}$ (Ado1.0mut) into the NAc of WT mice, we utilized fiber photometry to evaluate fluorescence signals indicative of extracellular adenosine (Fig. 2a and Supplementary Fig. 2a). Extracellular adenosine levels in the NAc rapidly increased after the transition from SWS or rapid eye movement sleep (REMS) to wakefulness. In contrast, the elevated extracellular adenosine levels at the wake transition showed a relatively slower decrease during SWS (Fig. 2b, c), rendering the fluorescence adenosine signal indistinguishable from that of Ado1.0mut at the transition to wakefulness or REMS (Supplementary Fig. 2b). Adenosine levels significantly increased for 3 h after treatment with 0.3 mg kg$^{-1}$ CNO when hM3Dq DREADD was expressed in astrocytes as compared to the vehicle control (Fig. 2d, e; Fig. 2d, $Q < 0.01$, false discovery rate with Benjamini-Hochberg adjustment; Fig. 2e, unpaired 2-tailed t-test, and Supplementary Fig. 2d) or neurons (Fig. 2f–h; Fig. 2g, $Q < 0.05$, false discovery rate with Benjamini-Hochberg adjustment; Fig. 2h, unpaired 2-tailed t-test). In contrast, after chemogenetic activation of astrocytes or under the vehicle control, no relevant Ado1.0mut signal was observed (Supplementary Fig. 2e, f). While astrocyte activation increased extracellular adenosine and induced sleep (Figs. 1 and 2), activation of NAc GABAergic medium spiny neurons did not induce sleep (Supplementary Fig. 3a–c;

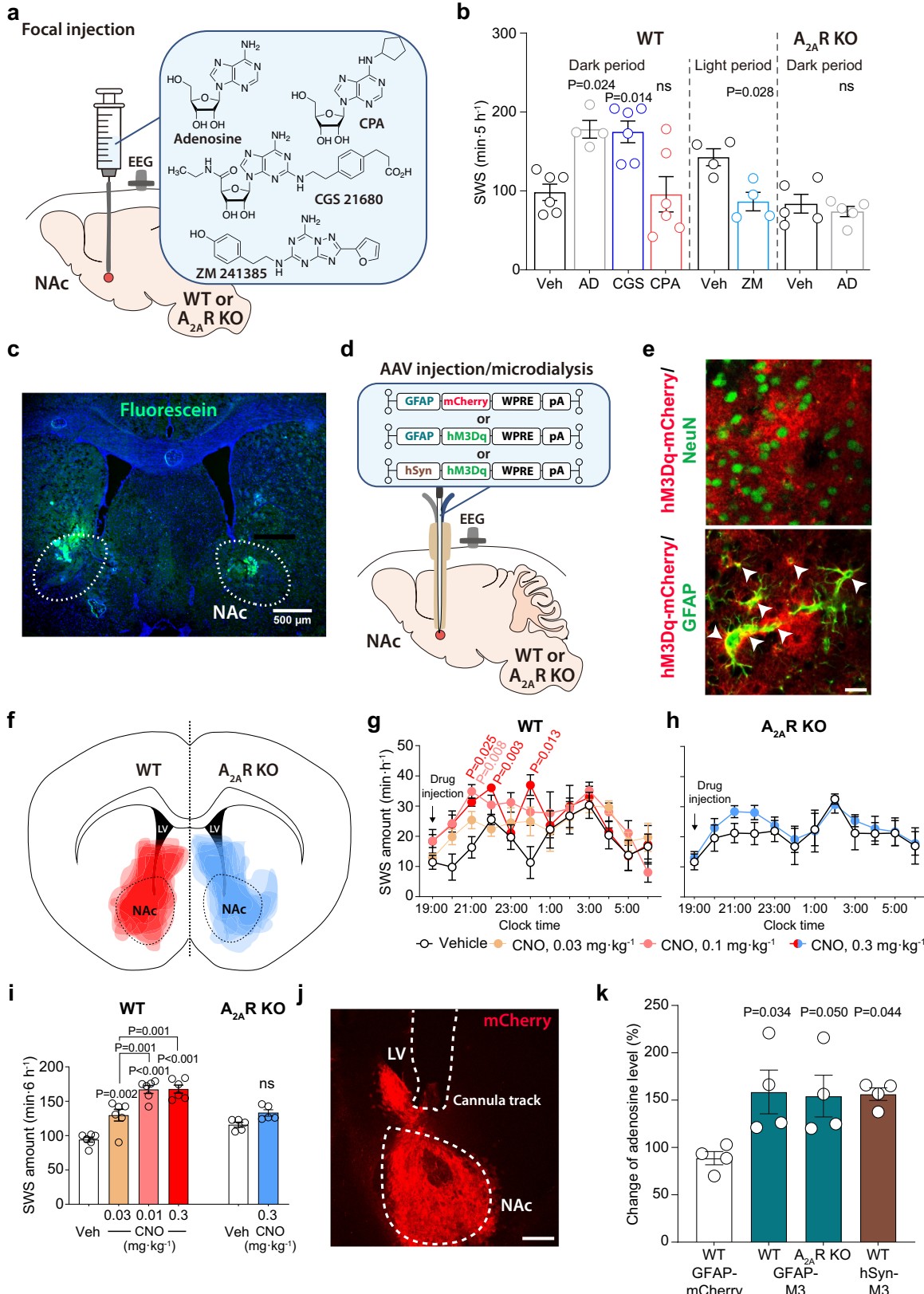

Supplementary Fig. 3b, 2-way repeated measures ANOVA-Bonferroni's multiple comparisons; Supplementary Fig. 3c, Mann-Whitney test), probably due to the opposing sleep/wake effects of NAc direct and indirect pathway neurons[11,17]. This suggests that astrocytes directly increase extracellular adenosine levels upon activation rather than stimulating neurons to increase extracellular adenosine.

We also reduced astrocyte calcium signaling by expressing the human plasma membrane calcium pump isoform type 2 (hPMCA2), which constitutively extrudes cytosolic calcium (Supplementary Fig. 4a)[18,19]. Extracellular adenosine levels in the NAc were significantly reduced after the transition from SWS to wakefulness and increased during REM sleep (Supplementary Fig. 4b, c, Q < 0.05, Wilcoxon

**Fig. 1 | Activation of NAc $A_{2A}R$ by focal injection of adenosine or stimulation of astrocytes induces SWS. a** Schematic of pharmacologic activation of NAc by focal injection of adenosine, CGS 21680, CPA, or ZM 241385 into freely behaving WT and $A_{2A}R$ KO mice, illustrated by Sara Kobayashi. **b** Total amount of SWS for 5 h after focal drug injection into the NAc. Data [left to right, n = 6 (Veh), 4 (AD), 6 (CGS), 6 (CPA), 4 (Veh), 4 (ZM), 5 (Veh), and 5 (AD) biologically independent animals in each group] are presented as mean ± SEM. Unpaired 2-tailed t-test compared with the vehicle injections. **c** Fluorescence image showing a representative example of the injection site with fluorescein located in the NAc. The experiment was independently repeated 2 times. **d** Schematic of AAV microinjection and placement of microdialysis probe in the NAc of WT or $A_{2A}R$ KO mice, illustrated by Sara Kobayashi. **e** Immunostaining for NeuN (upper panel) or GFAP (lower panel) together with mCherry in WT mice injected with an AAV expressing a hM3Dq DREADD/mCherry fusion protein under the GFAP promoter (AAV-GFAP-hM3DqDREADD). The experiment was independently repeated 6 (WT) and 5 ($A_{2A}$ KO) times. Scale bar: 20 μm. **f** Drawings of superimposed AAV-GFAP-hM3DqDREADD injection sites in the NAc core of WT (in red) and $A_{2A}R$ KO (in blue) mice are shown. Time course (**g, h**) and total amount (**i**) of SWS after chemogenetic stimulation of astrocytes in the NAc of WT (**g, i**) and $A_{2A}R$ KO (**h, i**) mice. Data [n = 6

(**g**, WT groups in **i**) and n = 5 (**h**, $A_{2A}R$ KO groups in **i**) biologically independent animals in each group] are presented as mean ± SEM. Unpaired 2-tailed t-test compared with vehicle (saline) injection. **j** Typical implantation site for the guide cannula and location of the microdialysis probe in the NAc. Immunostaining for mCherry indicates the AAV-infected area in the NAc. Scale bar: 400 μm. **k** Extracellular adenosine levels normalized to vehicle (saline) injections in the NAc of WT mice injected with AAV-GFAP-mCherry, AAV-GFAP-hM3DqDREADD or AAV-hSyn-hM3DqDREADD, $A_{2A}R$ KO mice injected with AAV-GFAP-hM3DqDREADD. Data (n = 4 biologically independent animals/group) are presented as mean ± SEM. Unpaired 2-tailed t-test compared with AAV-GFAP-mCherry-injected mice. Source data have been deposited in the Figshare database [https://doi.org/10.6084/m9.figshare.25468084]. Abbreviations: AAV adeno-associated virus, AD adenosine, $A_{2A}R$ adenosine $A_{2A}$ receptor, CNO clozapine N-oxide, DREADD designer receptors exclusively activated by designer drugs, EEG electroencephalogram, GFAP glial fibrillary acidic protein, hSyn human synapsin, KO knockout, LV lateral ventricle, NAc nucleus accumbens, NeuN neuronal nuclei, ns not significant, pA poly-adenylation signal, SEM standard error of the mean, SWS slow-wave sleep, Veh vehicle, WPRE woodchuck hepatitis virus posttranscriptional regulatory element, WT wild type.

signed-rank test). The sleep-wake behavior of mice with hPMCA2 expression in NAc astrocytes, however, was not changed (Supplementary Fig. 4d, e). The chemogenetic inhibition of NAc neurons (Supplementary Fig. 4f) showed a tendency to lower extracellular adenosine levels. A statistically significant reduction was observed at 3 h following CNO treatment compared to the control group injected with the vehicle (Supplementary Fig. 4g, h, unpaired 2-tailed t-test).

Overall, these results suggest that astrocytic and neuronal activity increases adenosine levels in the NAc. We, therefore, reasoned that the adenosine/NAc $A_{2A}R$ SWS regulatory circuit would be an ideal system to explore the potential of in-vivo $A_{2A}R$ optochemistry based on a photoactivatable PAM.

### Generation of photoactivatable $A_{2A}R$ PAM

The 2-nitroveratryl (Nv) group is among the most widely used photolabile-protecting groups (PPGs)[20]. We synthesized an Nv derivative of $A_{2A}RPAM$-1 by condensation of the carboxyl group of $A_{2A}RPAM$-1 with the hydroxyl group of 2-nitroveratrole alcohol (Supplementary Fig. 5a). Nv-$A_{2A}RPAM$-1 has strong absorption in the UV spectrum above 300 nm and thus, UV light at 365 nm can be used to photoactive Nv-$A_{2A}RPAM$-1, while avoiding strong UV light absorption of $A_{2A}RPAM$-1 (Supplementary Fig. 6a, b). UPLC analysis of Nv-$A_{2A}RPAM$-1 before and after UV light irradiation revealed incomplete photo-uncaging of Nv-$A_{2A}RPAM$-1 in water, and large amounts of Nv-$A_{2A}RPAM$-1 were detected even after 30 min of irradiation (Supplementary Fig. 6c), possibly due to the weak UV light absorption of $A_{2A}RPAM$-1 at 365 nm (see insert in Supplementary Fig. 6b), which may affect the photoreaction. In addition, irradiation with UV light leads to byproducts from deiodination and hydroxylation of $A_{2A}RPAM$-1, and is likely to damage biological tissues such as the brain in vivo.

We, therefore, developed a PPG, named A400, that is photoactivatable in the visible light spectrum above 400 nm. A400 is a water-soluble PPG derived from a previously developed coumarin-based caged compound[21]. The A400-$A_{2A}RPAM$-1 conjugate (opto$A_{2A}RPAM$-1), with a quantum yield of 0.521 ± 0.00717 (i.e., the number of $A_{2A}RPAM$-1 molecules produced per absorbed photon), was synthesized by condensation of the carboxyl group of $A_{2A}RPAM$-1 with the hydroxyl group of A400-2 followed by deprotection and salt formation (Supplementary Fig. 5b). The absorption maximum of opto$A_{2A}RPAM$-1 is at 415 nm and violet light at 405 nm can thus be used for opto$A_{2A}RPAM$-1 photoactivation (Fig. 3a, b). UPLC analysis of opto$A_{2A}RPAM$-1 before and after violet light irradiation showed complete photo-uncaging of a 200 μM solution of opto$A_{2A}RPAM$-1 in water after 30 s of irradiation (Fig. 3c, d). Moreover, treatment of $A_{2A}R$-expressing Chinese hamster ovary cells (CHO) with

opto$A_{2A}RPAM$-1 had no effect without violet light irradiation, and a significant increase in cAMP levels was observed after 10 s of light irradiation, reaching levels similar to $A_{2A}RPAM$-1 after 30 s (Fig. 3e). These in-vitro experiments indicate that opto$A_{2A}RPAM$-1 is a very effective photocaged derivative of $A_{2A}RPAM$-1 in aqueous solutions.

### In-vivo optodialysis of opto$A_{2A}RPAM$-1 into the NAc induces SWS

We next performed whole-cell patch-clamp electrophysiology of acutely prepared coronal brain slices (250 μm thick) containing the striatum with the NAc from WT and $A_{2A}R$ KO mice to investigate the effect of opto$A_{2A}RPAM$-1 photo-uncaging on the resting membrane potential (RMP) of NAc neurons (Fig. 4a). Based on their minimum current amplitude to generate action potentials (rheobase), we identified neurons with a low and high rheobase as indirect pathway neurons that express $A_{2A}Rs$ ($A_{2A}R^+$) and direct pathway neurons that do not express $A_{2A}Rs$ ($A_{2A}R^-$), respectively (Supplementary Fig. 7a, b) and verified that $A_{2A}RPAM$-1 increased the RMP of $A_{2A}R^+$ neurons, but not $A_{2A}R^-$ neurons, in brain slices perfused with artificial cerebrospinal fluid (aCSF) containing 200 μM $A_{2A}RPAM$-1 (Supplementary Fig. 7c). Adding adenosine to the slice preparation was unnecessary as the ex vivo adenosine levels were sufficient to enhance the RMB of NAc indirect pathway neurons by $A_{2A}RPAM$-1. The RMP of NAc $A_{2A}R^+$ indirect pathway neurons in brain slices perfused with aCSF containing 1 mg mL$^{-1}$ opto$A_{2A}RPAM$-1 was increased by stimulation with brief pulses (15 ms with 3-ms delay) of violet light (405 nm), whereas the RMP was not affected in the absence of violet light (Fig. 4a, c). In contrast, the RMP of $A_{2A}R^-$ direct pathway neurons decreased in the slice preparation by stimulation with violet light (Fig. 4b, c), likely due to lateral inhibition by $A_{2A}R$-expressing medium spiny neurons[22,23]. The RMP of NAc neurons in brain slices from $A_{2A}R$ KO mice perfused with aCSF containing 1 mg mL$^{-1}$ opto$A_{2A}RPAM$-1 did not change after stimulation with violet light (Fig. 4c). These results suggest that photo-uncaging of opto$A_{2A}RPAM$-1 in the NAc results in selective depolarization of $A_{2A}R$-expressing indirect pathway neurons.

To investigate the effects of opto$A_{2A}RPAM$-1 photolysis in the NAc on sleep/wake behavior in WT mice, we first used an optodialysis probe that combines optical and microdialysis probes in one unit[24] to infuse 10 nmol min$^{-1}$ opto$A_{2A}RPAM$-1 into the NAc of WT mice during the dark period (20:00–8:00) and assessed EEG and EMG activity (Fig. 4d, e). Compared with vehicle (Ringer's solution) infusion, 1-h illumination of the NAc with 15-ms pulses (3-ms delay) of violet light significantly increased SWS for 1 h after photostimulation (Fig. 4f, F(20,120) = 1.467, P = 0.1060, 2-way repeated measures ANOVA-Tukey test). $A_{2A}RPAM$-1 induces SWS in mice for several hours after intraperitoneal injection[25].

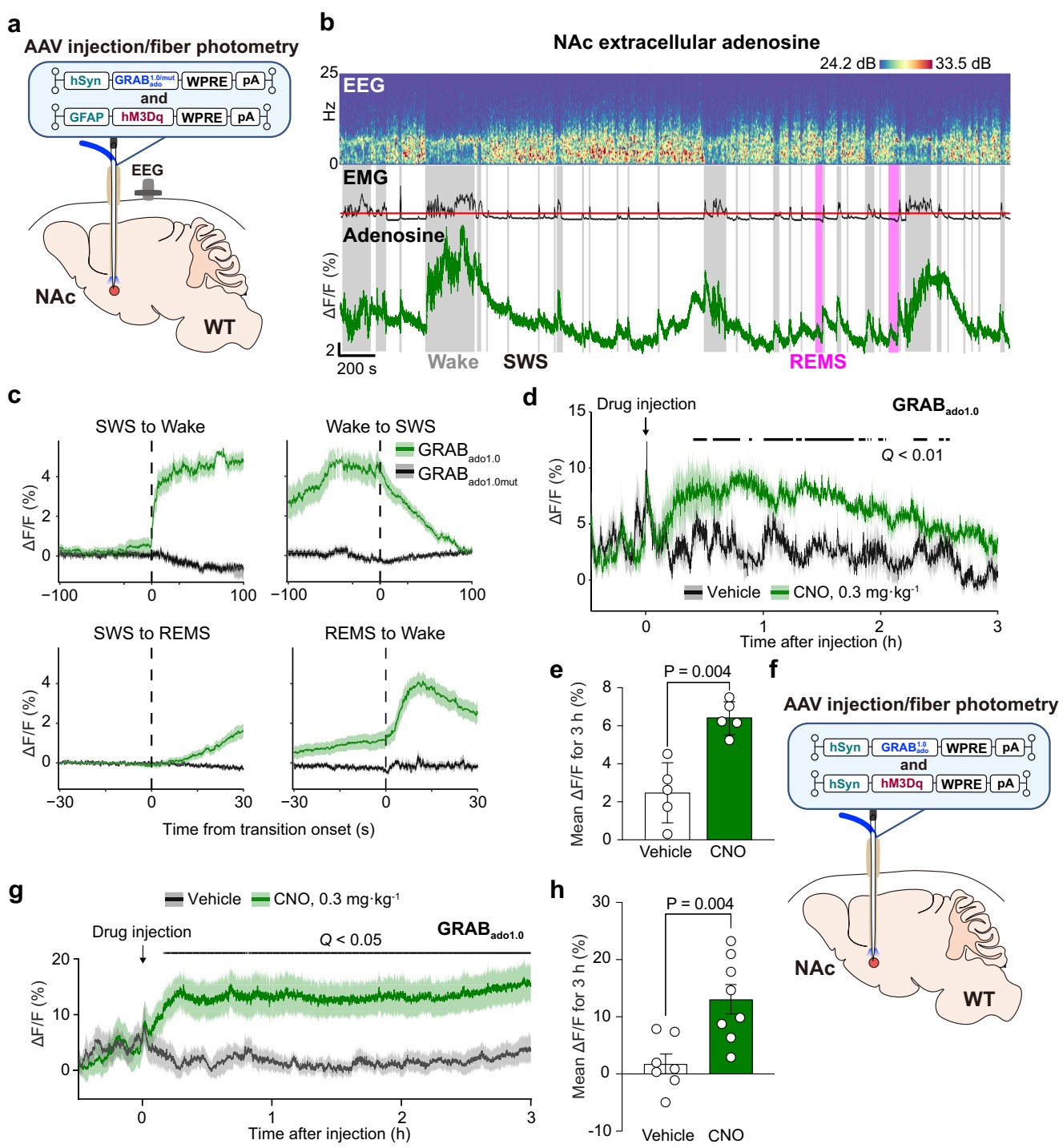

We, therefore, calculated total SWS for 5 h after light stimulation; SWS only tended to change in opposite directions and the changes were not statistically significant (Fig. 4g). REMS was not affected compared with that in the control experiments without light irradiation (Fig. 4h, i). These results suggest that photolysis of optoA$_{2A}$RPAM-1 in the NAc of WT mice selectively induces SWS.

## Systemic administration of optoA$_{2A}$RPAM-2, but not optoA$_{2A}$RPAM-1, together with NAc photoirradiation induces SWS

As previously reported, A$_{2A}$RPAM-1 induces SWS after intraperitoneal injection in mice[9]. Therefore, we next investigated whether systemic administration of optoA$_{2A}$RPAM-1 and photoirradiation of the NAc

could induce SWS. We implanted optical fibers bilaterally into the NAc of WT mice and analyzed EEG and EMG recordings made after intraperitoneal injections of 10 mL kg$^{-1}$ saline or 150 mg kg$^{-1}$ optoA$_{2A}$RPAM-1 at 21:00 to measure sleep (Fig. 5a, b). We stimulated the NAc in vivo with brief 15-ms pulses (3-ms delay) of violet light for 1 h to release A$_{2A}$RPAM-1 locally in the NAc. We observed no significant changes in the hourly or 5-h SWS/REMS amounts in WT mice (Fig. 5c–f; Fig. 5c, $F_{(20,100)} = 0.7607$, $P = 0.7530$, 2-way repeated measures ANOVA-Tukey test; Fig. 5d, $F_{(2,10)} = 1.949$, $P = 0.1928$, one way ANOVA-Tukey test; Fig. 5e, $F_{(20,100)} = 1.388$, $P = 0.1465$, 2-way repeated measures ANOVA-Tukey test; Fig. 5f, $F_{(2,10)} = 0.4223$, $P = 0.6667$, one-way ANOVA-Tukey test), however, indicating that sufficient amounts of optoA$_{2A}$RPAM-1 did not cross the BBB. Therefore, we measured the

**Fig. 2 | Activation of astrocytes or neurons in the NAc increases extracellular adenosine levels. a** Schematic of microinjection of AAVs expressing neuronal Ado1.0 and astrocytic hM3Dq DREADD and optic fiber placement in the NAc of WT mice, illustrated by Sara Kobayashi. **b** Typical examples of EEG, EMG, and adenosine signals for 1 h in the NAc of a WT mouse. The EMG trace is shown as root mean square and the red line indicates level 5 activity. **c** Mean NAc adenosine levels before and after each state transition. Data [Transitions examined over 5 independent experiments: Wake-SWS, n = 30 (Ado1.0) or 26 (Ado1.0mut); SWS-Wake, n = 30 (Ado1.0) or 25 (Ado1.0mut); SWS-REMS, n = 30 (Ado1.0 or Ado1.0mut); REMS-SWS, n = 27 (Ado1.0) or 26 (Ado1.0mut)] are presented as mean ± SEM (shaded areas). Time course (**d**) and mean (**e**) adenosine levels after chemogenetic stimulation of astrocytes in the NAc of WT. Data (n = 5 biologically independent animals/group) are presented as mean ± SEM [shaded areas in (**d**) or error bars in (**e**)]. **d** Horizontal bars indicate false discovery rate Q. **e** Unpaired 2-tailed t-test compared with vehicle (saline) injection. **f** Schematic of microinjection of AAVs expressing neuronal Ado1.0 and hM3Dq DREADD and optic fiber placement in the NAc of WT mice, illustrated by Sara Kobayashi. Time course (**g**) and mean (**h**) adenosine levels after chemogenetic stimulation of neurons in the NAc of WT. Data [**g**, **h**, n = 7 (vehicle) and n = 8 (CNO) biologically independent animals in each group] are presented as mean ± SEM [shaded areas in (**g**) or error bars in (**h**)]. **g** Horizontal bar indicates false discovery rate Q. **h** Unpaired 2-tailed t-test compared with vehicle (saline) injection. Source data have been deposited in the Figshare database [https://doi.org/10.6084/m9.figshare.25468084]. Abbreviations: AAV adeno-associated virus, CNO clozapine N-oxide, DREADD designer receptors exclusively activated by designer drugs, EEG electroencephalogram, EMG electromyogram, GFAP glial fibrillary acidic protein, GRAB$_{Ado1.0/mut}$ G protein-coupled receptor-activation-based adenosine sensor Ado1.0 or non-binding mutant Ado1.0mut, hSyn human synapsin, NAc nucleus accumbens, pA polyadenylation signal, REMS rapid eye movement sleep, SEM standard error of the mean, SWS slow-wave sleep, WPRE woodchuck hepatitis virus posttranscriptional regulatory element, WT wild type.

brain concentration of A$_{2A}$RPAM-1 30 min after intraperitoneal administration of A$_{2A}$RPAM-1 or optoA$_{2A}$RPAM-1 using UPLC coupled with tandem mass spectrometry (UPLC-MS/MS). Whereas significant amounts of A$_{2A}$RPAM-1 were detected in the brain when A$_{2A}$RPAM-1 was injected intraperitoneally, optoA$_{2A}$RPAM-1 injected intraperitoneally followed by exposure of homogenized brain samples to light was not detected as A$_{2A}$RPAM-1in the brain, suggesting that optoA$_{2A}$RPAM-1 cannot pass the BBB to induce optochemical sleep (Fig. 5g, h). The BBB is a semipermeable barrier that separates the extracellular fluid surrounding the brain from circulating blood, and pharmacologically active compounds must strike a balance between hydrophilicity and hydrophobicity[26]. Thus, we also measured the A$_{2A}$RPAM-1 brain concentration at 30 min after intraperitoneal administration of the more hydrophobic optoA$_{2A}$RPAM-2, a precursor for the synthesis of optoA$_{2A}$RPAM-1, followed by exposure of homogenized brain samples to light by UPLC-MS/MS and, surprisingly, detected A$_{2A}$RPAM-2 in the brain samples (Fig. 5g, h).

We, therefore, investigated whether systemic administration of optoA$_{2A}$RPAM-2, which has an absorption maximum at a slightly shorter wavelength ($\lambda_{max}$ = 407 nm) than optoA$_{2A}$RPAM-1 (Fig. 3b), and photoirradiation of the NAc with violet light could induce SWS (Fig. 6a). EEG and EMG recordings made after intraperitoneal injections of 10 mL kg$^{-1}$ vehicle or 150 mg kg$^{-1}$ optoA$_{2A}$RPAM-2 at 21:00 showed that SWS was significantly increased for 5 h after 1-h illumination with violet light compared with the injection of vehicle or optoA$_{2A}$RPAM-2 without photoirradiation (Fig. 6b, c, F(20,120) = 2.489, P = 0.0012, 2-way repeated measures ANOVA-Tukey test). Accordingly, the total SWS amount for 5 h was significantly increased compared with that following the injection vehicle or optoA$_{2A}$RPAM-2 without photoirradiation (Fig. 6d). By contrast, we observed no significant changes in REMS (Fig. 6e, f). Systemic administration of optoA$_{2A}$RPAM-2 together with NAc photoirradiation significantly increased the mean SWS episode number for 5 h (Supplementary Fig. 8a, one-way repeated measures ANOVA-Bonferroni's multiple comparisons, F(2,12) = 9.386, P = 0.0035), whereas focal NAc photouncaging of optoA$_{2A}$RPAM-2 did not affect the mean REMS episode number (Supplementary Fig. 8b). A significantly increased number of SWS episodes with a duration from 20 to 50 s was observed for 5 h after optoallosteric NAc A$_{2A}$R stimulation (Supplementary Fig. 8c, 2-way repeated measures ANOVA-Bonferroni's multiple comparisons, F(12,72) = 3.269, P = 0.0008). These results suggest that photoactivation of the A$_{2A}$RPAM-1 in the NAc of mice induces sleep by increasing the RMP of indirect pathway neurons expressing A$_{2A}$Rs. We did not analyze other behaviors after photoactivation of the A$_{2A}$RPAM-1 in the NAc because A$_{2A}$RPAM-1, likely due to its sleep-inducing properties, strongly suppresses behaviors such as risk-taking in an open field test in WT as well as in microtubule-associated protein 6

(MAP6) KO mice (Supplementary Fig. 9, unpaired Mann-Whitney test), a genetic mouse model of schizophrenia/psychosis[10,27].

## NAc A$_{2A}$Rs are required for the induction of SWS by a photo-activatable A$_{2A}$RPAM-1

We next investigated whether NAc A$_{2A}$Rs are required for optochemical control of SWS by a photoactivatable A$_{2A}$RPAM-1. First, 150 mg kg$^{-1}$ optoA$_{2A}$RPAM-2 was intraperitoneally administered to A$_{2A}$R KO mice and EEG/EMG activity was assessed after the NAc was illuminated for 1 h with 15-ms pulses (3-ms delay) of violet light (Fig. 6g). We observed no changes in SWS or REMS after selective photolysis of optoA$_{2A}$RPAM-2 in the NAc of A$_{2A}$R KO mice compared with the vehicle control (Fig. 6h and Supplementary Fig. 10a–c; multiple unpaired t-tests).

Moreover, whole-cell recordings of ventrolateral preoptic area (VLPO) neurons in rat brain slices revealed that A$_{2A}$Rs on VLPO neurons are also involved in initiating sleep[28]. We next tested whether SWS can be induced by systemic administration of 150 mg kg$^{-1}$ optoA$_{2A}$RPAM-2 and photoirradiation of VLPO neurons with violet light (Fig. 6i). No changes in SWS or REMS were observed after optoA$_{2A}$RPAM-2 photolysis in the VLPO of WT mice compared with the vehicle control (Fig. 6j, and Supplementary Fig. 10d–f; multiple unpaired t-tests). These results suggest that selective allosteric modulation of NAc A$_{2A}$Rs is necessary for SWS induction.

## Discussion

In this study, we have developed a brain-permeable A$_{2A}$R PAM that can be rapidly photoactivated with visible light ($\lambda$ > 400 nm) and optochemically induced sleep in freely behaving mice by enhancing A$_{2A}$R signaling of NAc indirect pathway neurons. Our approach holds great potential for generating photoactivatable compounds for virtually any preclinical and translational biologic target. In the future, optochemistry may offer promising avenues for alleviating symptoms, treating diseases, and improving quality of life while minimizing unintended drug actions.

The NAc is a key area where sleep and behavioral responses to motivational stimuli are reciprocally regulated. Adenosine may play a key role in SWS control in the NAc because activation of A$_{2A}$R-expressing NAc indirect pathway neurons induces sleep. Moreover, caffeine, a natural adenosine receptor antagonist, produces its arousal effect through NAc A$_{2A}$R in mice[12] and pathologically elevated adenosine levels in the NAc promote sleep[29]. No direct evidence has been presented, however, for the physiologic importance of adenosine in the NAc for the regulation of sleep. Allosteric modulators only exert their effects where and when the orthosteric ligand such as adenosine is bound to its target receptor. Optoallosteric modulation of A$_{2A}$R signaling in the NAc increased SWS, suggesting that sufficient levels of

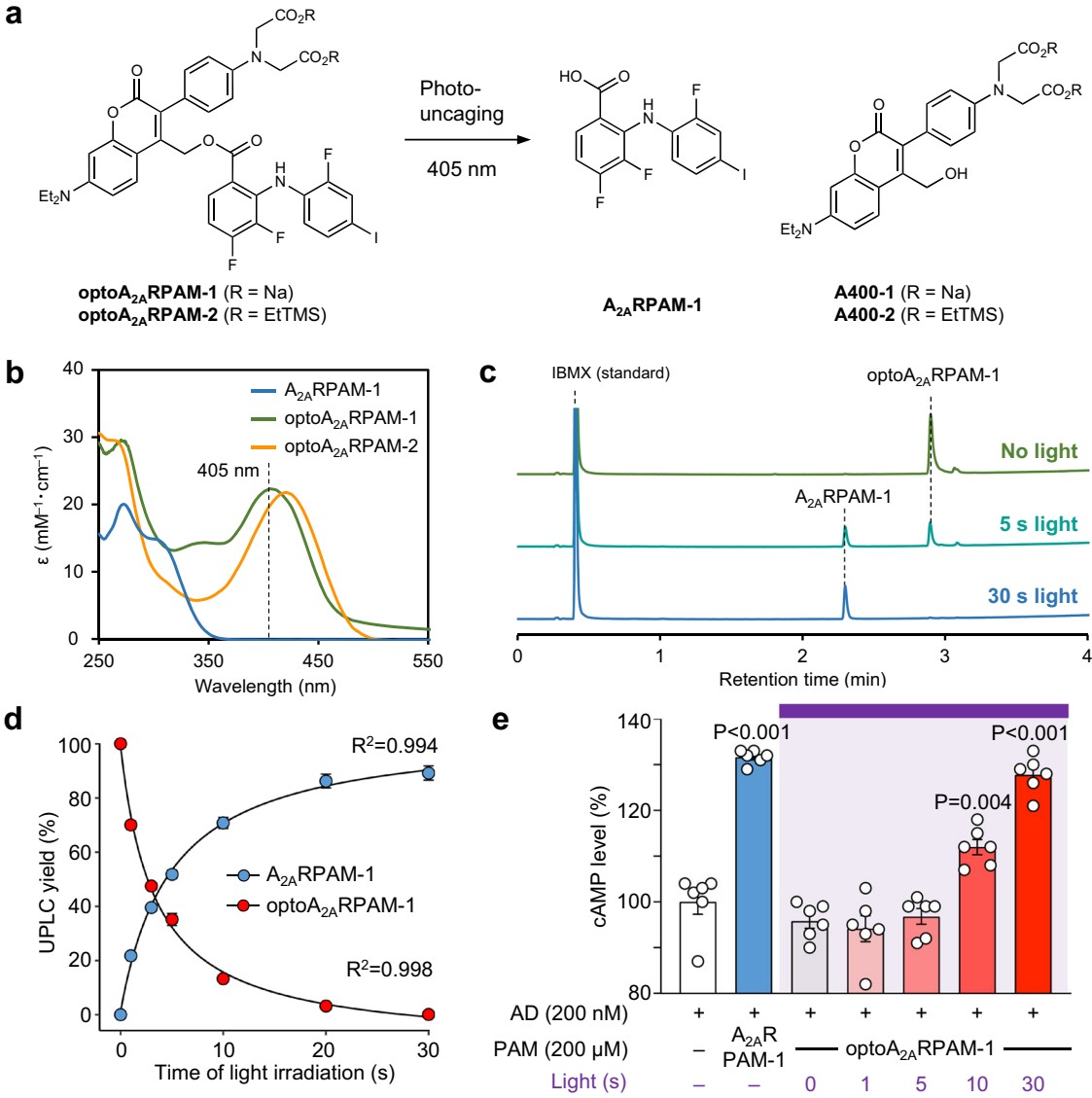

**Fig. 3 | Generation of photoactivatable optoA₂AR PAM. a** Schematic of photo-uncaging of optoA₂ARPAM-1 by visible light (405 nm). **b** UV–visible spectrum of A₂ARPAM-1, optoA₂ARPAM-1, and optoA₂ARPAM-2. **c**, **d** UPLC analysis of optoA₂ARPAM-1 photo-uncaging with violet light. UPLC traces (**c**) and quantification (**d**). **d** Experiments were performed in triplicate for each light irradiation duration, and data (n = 3 independent experiments/time point) are presented as mean ± SEM with R squared value for 2-phase exponential curve fit. **e** Time-dependent changes of cAMP levels in mA₂AR-expressing Chinese hamster ovary cells after treatment with adenosine, A₂ARPAM-1, and optoA₂ARPAM-1 and various durations of light exposure. Experiments were performed in triplicate wells for each condition, and data (n = 6 independent experiments/group) are presented as mean ± SEM. Unpaired 2-tailed t-test compared to adenosine control. Source data have been deposited in the Figshare database [https://doi.org/10.6084/m9.figshare.25468084]. Abbreviations: AD adenosine, EtTMS trimethylsilylethanol derivative, IBMX isobutylmethylxanthine, Na sodium, PAM positive allosteric modulator, SEM standard error of the mean, UPLC ultra-high-performance liquid chromatography, UV ultraviolet.

extracellular adenosine are available in the NAc under physiologic conditions to promote SWS. Sleep-regulating adenosine in the NAc may originate from astrocytes and neurons, but the exact source of the adenosine remains unclear[15,30–32]. Our data suggest that adenosine is released from astrocytes and neurons because chemogenetically induced activity in NAc astrocytes and neurons increases extracellular adenosine. Gi- and Gq-GPCR signaling in astrocytes is thought to increase Ca²⁺ activity via IP₃-dependent release of intracellular Ca²⁺[33–36], but chemogenetic hM3Dq DREADD activation of cortical astrocytes leads to a long-lasting silent state of Ca²⁺ dynamics after an initial short period of Ca²⁺ activity[37]. The signaling cascades, possibly Ca²⁺-independent, that lead to sustained adenosine release from NAc astrocytes upon hM3Dq DREADD stimulation, therefore, remains unclear.

The optochemically mediated sleep-promoting effect of A₂AR PAMs provides proof of concept for brain responses induced by optochemistry and thus opens the door for the potential therapeutic use of these chemicals for treating diseases. In addition to spatial control by local irradiation with light, A₂AR-PAM optochemistry uses allosteric modulation with which physiologic specificity can be achieved more easily than with classical agonist and antagonist drugs. Thus, A₂AR-PAM optochemistry could provide certain patients with effective and safe treatment for various diseases, such as treatment-resistant insomnia, also known as chronic insomnia, for which most available therapeutic options are unsatisfactory, leaving patients desperate for alternative therapies. Moreover, A₂AR PAM optochemistry may also represent a potential therapeutic approach to neuroinflammation. Adenosine is present in high concentrations in areas of

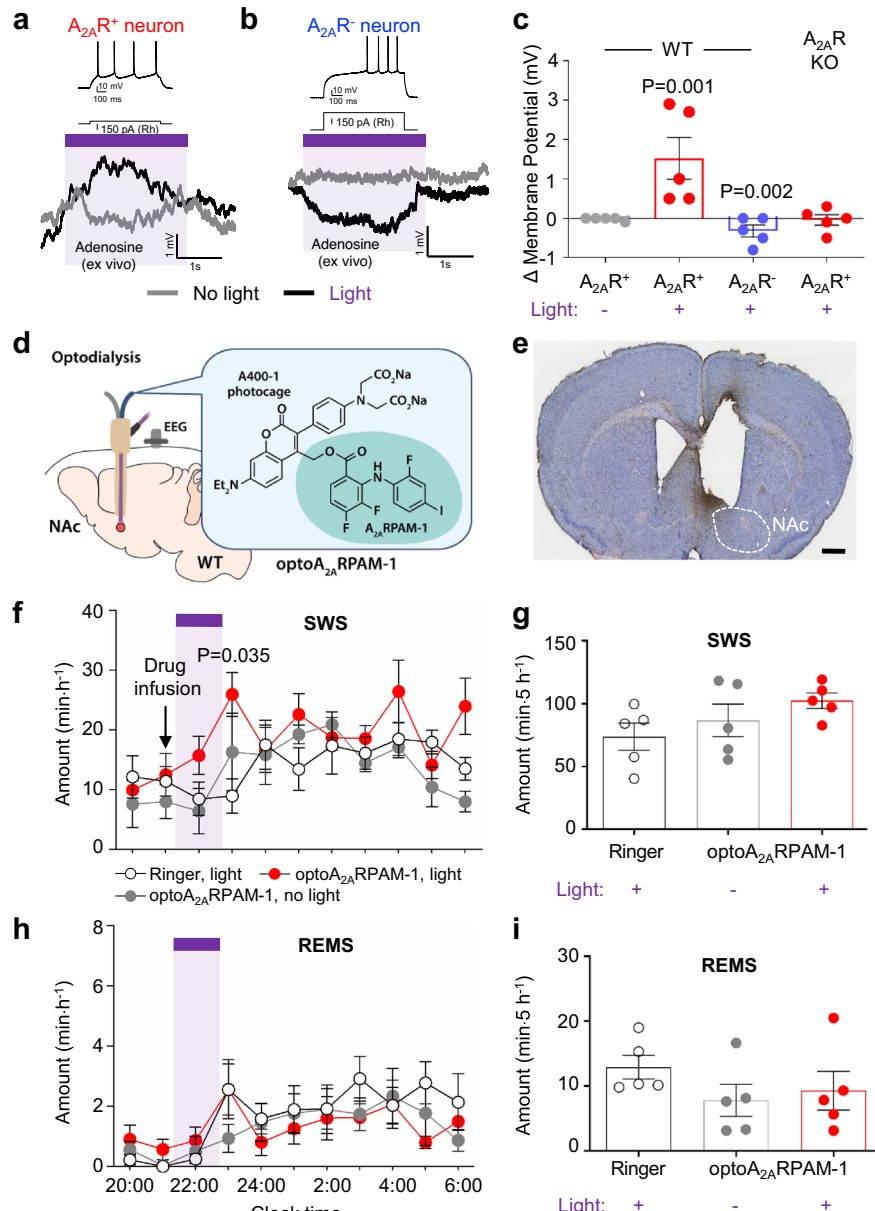

**Fig. 4 | In-vivo optodialysis of optoA$_{2A}$RPAM-1 into the NAc.** Resting membrane potential of NAc A$_{2A}$R$^+$ (**a**) and A$_{2A}$R$^-$ (**b**) neurons after treatment with optoA$_{2A}$RPAM-1 and illumination with violet light (405 nm). The rheobase (Rh) was used to identify A$_{2A}$R$^+$ (low Rh) or A$_{2A}$R$^-$ (high Rh) neurons (upper traces in a and b). **c** Changes in the membrane potential of NAc A$_{2A}$R$^+$ and A$_{2A}$R$^-$ neurons in WT mice and NAc A$_{2A}$R$^+$ in A$_{2A}$R KO mice by optoallosteric activation with optoA$_{2A}$RPAM-1. Data (n = 5 biologically independent animals/group) are presented as mean ± SEM. Unpaired 2-tailed t-test compared with optoA$_{2A}$RPAM-1 treatment without light. **d** Schematic of optoA$_{2A}$RPAM-1 optodialysis into the NAc of WT mice, illustrated by Sara Kobayashi. **e** Histologic verification of typical cannula placement in the mouse NAc. The experiment was independently repeated 4 times. Scale bar: 1 mm. Time course (**f, h**) and total amount (**g, i**) of SWS and REMS after optoA$_{2A}$RPAM-1 optodialysis into the NAc of WT mice. **f, h** The purple bar indicates 1-h light illumination. Data (n = 5 biologically independent animals/group) are presented as mean ± SEM. Unpaired 2-tailed t-test compared with the infusion of Ringer's solution. Source data have been deposited in the Figshare database [https://doi.org/10.6084/m9.figshare.25468084]. Abbreviations: A$_{2A}$R adenosine A$_{2A}$ receptor, EEG electroencephalogram, KO knockout, NAc nucleus accumbens, REMS rapid eye movement sleep, SEM standard error of the mean, SWS slow-wave sleep, WT wild type.

inflammation due to cell activation and breakdown[38–40] and A$_{2A}$Rs are responsible for the anti-inflammatory effects of adenosine[41,42]. A$_{2A}$R PAM optochemistry may also alleviate symptoms of psychotic disorders such as schizophrenia. Psychotic symptoms such as delusions are caused by impaired discrimination of environmental stimuli. Previous reports showed that NAc A$_{2A}$Rs are involved in the impairment of discrimination learning, working memory, and psychomotor activity associated with schizophrenia[43,44]. Moreover, 40-Hz light flickering, recognized by the United States Food and Drug Administration as a prospective therapy for Alzheimer's disease[45,46], elevates extracellular

adenosine levels in several brain regions that control important physiological processes, including the NAc[19]. Combining light flickering to induce ligand release with PAM optochemistry to enhance receptor activity may pave the way for innovative light-based therapeutic strategies to treat sleep and various brain disorders in the future.

Drug delivery across the BBB is a common challenge in the development of therapeutics for the brain. Photocaging of drugs that are initially brain-permeable using PPGs may impede drug delivery to the central nervous system. Whereas relatively lipophilic monocarboxylic acids such as A$_{2A}$RPAM-1 can easily pass the BBB by passive diffusion or

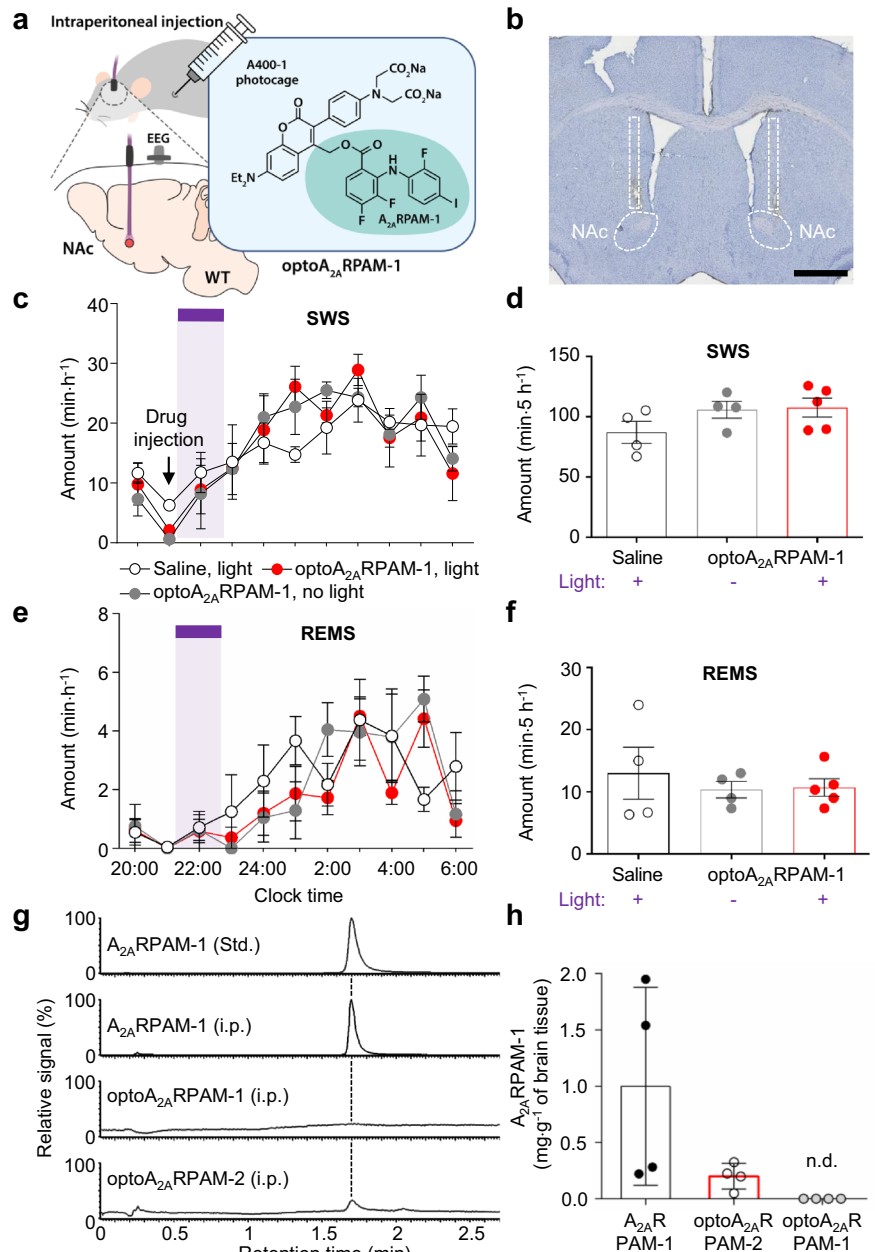

**Fig. 5 | Systemic administration of optoA$_{2A}$RPAM-1 together with NAc photo-irradiation. a** Schematic diagram of systemic administration (intraperitoneal) of 150 mg kg⁻¹ optoA$_{2A}$RPAM-1 and NAc photoirradiation with violet light (405 nm) for 1 h, illustrated by Sara Kobayashi. **b** Histologic verification of typical optic cannula placement in the mouse NAc. The experiment was independently repeated 4 times. Scale bar: 1 mm. Time course (**c**, **e**) and total amount (**d**, **f**) of SWS and REMS after optoallosteric NAc activation with optoA$_{2A}$RPAM-1. **c**, **e** The purple bar indicates 1-h light illumination. Data [n = 4 (saline), 4 (optoA$_{2A}$RPAM-1/no light) and n = 5 (optoA$_{2A}$RPAM-1/light) biologically independent animals in each group] are presented as mean ± SEM. **g**, **h** UPLC-MS/MS analysis of brain samples from mice

injected intraperitoneally with 75 mg kg⁻¹ A$_{2A}$RPAM-1, 150 mg kg⁻¹ optoA$_{2A}$RPAM-1, or 150 mg kg⁻¹ optoA$_{2A}$RPAM-2. Single ion signal (**g**) and total brain concentrations (**h**) of A$_{2A}$RPAM-1 in the brain samples exposed to violet light. Data (n = 4 biologically independent animals/group) are presented as mean ± SEM. Source data have been deposited in the Figshare database [https://doi.org/10.6084/m9.figshare.25468084]. Abbreviations: A$_{2A}$R adenosine A$_{2A}$ receptor, EEG electro-encephalogram, i.p. intraperitoneal, MS/MS tandem mass spectrometry, NAc nucleus accumbens, n.d. not detectable, REMS rapid eye movement sleep, SEM standard error of the mean, Std. standard, SWS slow-wave sleep, UPLC ultra-high-performance liquid chromatography, WT wild type.

via a monocarboxylate transport system[47], the highly water-soluble dicarboxylic acid optoA$_{2A}$RPAM-1 is unable to pass the BBB after intraperitoneal administration and thus cannot optochemically induce sleep. We, therefore, exploited the increased lipophilicity of optoA$_{2A}$RPAM-2, a precursor for the synthesis of optoA$_{2A}$RPAM-1, to achieve appropriate brain permeability and optochemical sleep induction.

Finally, the brain is by far the most difficult organ in the human body to irradiate with light because of the opacity of bone tissue. Therefore, light delivery to the brain currently requires an invasive

surgical procedure. Nevertheless, the use of light in the brain has gained attention with the advent of optogenetics, and novel ways to deliver light into the brain of model organisms such as mice have been established[48–51]. Given the great potential of optogenetics and opto-chemistry for the treatment of neurologic diseases, novel approaches to light transmission, such as near-infrared-light activatable nanoparticles[52], could overcome the major limitation of optochemistry in the brain.

Taken together, our experiments demonstrate the potential of in-vivo A$_{2A}$R PAM optochemistry for treating neurologic disorders. The

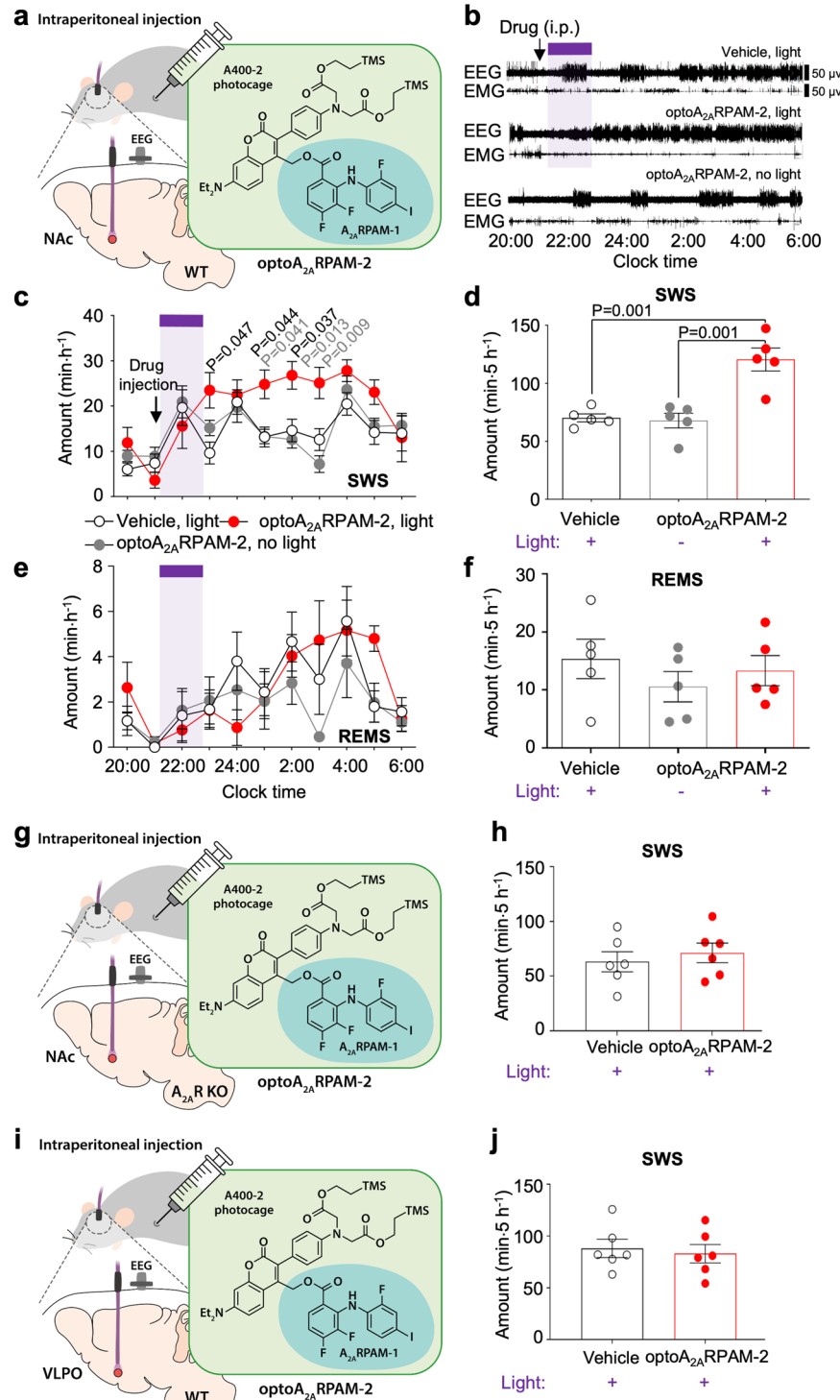

**Fig. 6 | Systemic administration of optoA$_{2A}$RPAM-2 together with photo-irradiation of the NAc in WT mice or A$_{2A}$R KO mice and the VLPO in WT mice.**
**a** Schematic diagram of systemic administration (intraperitoneal injection) of 150 mg kg$^{-1}$ optoA$_{2A}$RPAM-2 in WT mice and NAc photoirradiation with violet light (405 nm) for 1 h, illustrated by Sara Kobayashi. **b** Typical examples of EEG and EMG after administration of vehicle or optoA$_{2A}$RPAM-2 in a WT mouse and NAc pho-toirradiation. Time course (**c**, **e**) and total amount (**d**, **f**) of SWS and REMS after optoallosteric NAc activation with optoA$_{2A}$RPAM-2. Data (n = 5 biologically inde-pendent animals/group) are presented as mean ± SEM. Bonferroni t-test compared with vehicle (black font) or optoA$_{2A}$RPAM-2/no light (gray font). **c**, **e** The purple bar indicates 1-h light illumination. Schematic diagrams of systemic administration

(intraperitoneal injection) of 150 mg kg$^{-1}$ optoA$_{2A}$RPAM-2 in A$_{2A}$R KO or WT mice and NAc (**g**) or VLPO (**i**), respectively, and photoirradiation with violet light (405 nm) for 1 h, illustrated by Sara Kobayashi. Total amount of SWS after optoallosteric NAc (**h**) or VLPO (**j**) activation with optoA$_{2A}$RPAM-2. Data (n = 6 biologically independent animals/group) are presented as mean ± SEM. Source data have been deposited in the Figshare database [https://doi.org/10.6084/m9.figshare.25468084]. Abbreviations: A$_{2A}$R adenosine A$_{2A}$ receptor, EEG electro-encephalogram, EMG electromyogram, i.p. intraperitoneal, KO knockout, NAc nucleus accumbens, REMS rapid eye movement sleep, SEM standard error of the mean, SWS slow-wave sleep, VLPO ventrolateral preoptic area, WT wild type.

utilization of BBB-permeable optoA$_{2A}$R PAMs provides an advantage over conventional neuropharmacologic approaches through the high physiologic specificity of allosteric modulation and spatial control by light irradiation.

# Methods

## Chemistry

All commercially available chemicals and solvents were used without further purification. In general, reaction mixtures were magnetically stirred at the respective temperatures under argon atmosphere. Reactions were monitored by thin-layer chromatography (TLC). TLC and preparative TLC were carried out on silica gel plates (Kieselgel 60 F$_{254}$ [0.25 mm and 0.50 mm], Merck), in which existing compounds were visualized by UV light (254 nm) and stained with phosphomolybdic acid in an aqueous sulfuric acid solution and ninhydrin in an ethanol solution followed by heating. Flush column chromatography was performed on silica gel or C18-reverse-phase silica gel (CHROMATOREX PSQ60B, 40–50 µm, spherical, neutral; Fuji Silysia Chemical). $^1$H and $^{13}$C NMR spectra were recorded on a JEOL JNM-ECS 400 instrument ($^1$H: 400 MHz; $^{13}$C: 100 MHz). Chemical shifts are shown in parts per million (ppm) using tetramethylsilane ($\delta$ = 0.00 ppm) and metanol-$d_4$ (CD$_3$OD, $\delta$ = 3.31 ppm) as the internal standards for $^1$H NMR and using chloroform-$d$ (CDCl$_3$, $\delta$ = 77.16 ppm) and dimethylsulfoxide (DMSO)-$d_6$ ($\delta$ = 39.52 ppm) as the internal standards for $^{13}$C NMR spectroscopy. Signal patterns are indicated as br = broad; s = singlet; d = doublet; t = triplet, q = quintet, m = multiplet. Infrared spectra (IR) were recorded with a JASCO FT/IR-4100 spectrophotometer. Mass spectra were measured with a JEOL JMS-T100LP spectrometer. The purity and photolysis efficiency of the compounds were determined by analytical UPLC. Analytical UPLC was performed on a Waters ACQUITY UPLC system equipped with an ACQUITY UPLC BEH C18 (1.7 µm, 50 mm × 2.1 mm; Waters) column with a temperature of 40 °C and photodiode array detection at 254 nm. A graded 0.01% (v/v) acetic acid/acetonitrile mobile phase was used for UPLC separation.

## Synthesis of Nv-A$_{2A}$RPAM-1 (Supplementary Fig. 5a)

**4,5-Dimethoxy-2-nitrobenzyl 3,4-difluoro-2-((2-fluoro-4-iodophenyl)amino)benzoate (Nv-A$_{2A}$RPAM-1).** 1-[Bis(dimethylamino)methylene]-1*H*-1,2,3-triazolo[4,5-b]pyridinium 3-oxide hexafluorophosphate (HATU, 1.31 g, 3.44 mmol) and triethylamine (1.6 mL, 11.5 mmol) were added to a solution of 3,4-difluoro-2-((2-fluoro-4-iodophenyl)amino)benzoic acid (A$_{2A}$RPAM-1, 451 mg, 1.15 mmol) and (4,5-dimethoxy-2-nitrophenyl)methanol (**S1**, 489 mg, 2.29 mmol) in *N,N*-dimethylformamide (5 mL) and the mixture was stirred at room temperature for 5 h in the dark. The reaction mixture was diluted with aqueous saturated ammonium chloride solution (150 mL) and extracted with chloroform. The combined organic layer was washed with brine, dried over sodium sulfate, and concentrated in vacuo. The residue was purified by column chromatography on silica gel (eluent: hexane/ethyl acetate = 90:10–0:100 v/v) and recrystallized from chloroform to obtain 4,5-dimethoxy-2-nitrobenzyl 3,4-difluoro-2-((2-fluoro-4-iodophenyl)amino)benzoate (**Nv-A$_{2A}$RPAM-1**, 118 mg, 17 %); IR (NaCl): 3316, 2939, 2850, 1696, 1581, 1520, 1330, 1276, 1221, 1068, 912, 875, 732 cm$^{-1}$; $^1$H NMR (CDCl$_3$) $\delta$ = 8.97 (s, 1H), 7.84 (ddd, *J* = 9.2, 6.0, 2.3 Hz, 1H), 7.75 (s, 1H), 7.41 (dd, *J* = 10.0, 2.3 Hz, 1H), 7.34 (ddd, *J* = 8.2, 1.4 Hz, 1H), 7.01 (s, 1H), 6.78 (ddd, *J* = 9.2, 6.9 Hz, 1H), 6.70 (ddd, *J* = 8.7, 5.5 Hz, 1H), 5.72 (s, 2H), 3.97 (s, 3H), 3.94 (s, 3H); $^{13}$C NMR (CDCl$_3$) $\delta$ = 166.5 (d, $J_{C,F}$ = 2.9 Hz), 154.6 (dd, $J_{C,F}$ = 255.9, 11.5 Hz), 154.4 (d, $J_{C,F}$ = 251.1 Hz), 153.4, 148.8, 142.1 (dd, $J_{C,F}$ = 251.1, 14.4 Hz), 140.5, 136.5 (dd, $J_{C,F}$ = 7.7, 2.9 Hz), 133.1 (d, $J_{C,F}$ = 3.8 Hz), 129.8 (dd, $J_{C,F}$ = 12.0, 2.9 Hz), 127.4 (dd, $J_{C,F}$ = 9.6, 3.8 Hz), 126.1, 124.6 (d, $J_{C,F}$ = 22.0 Hz), 122.3 (d, $J_{C,F}$ = 5.8 Hz), 113.6, 111.3, 108.55, 108.52 (d, $J_{C,F}$ = 18.2 Hz), 84.2 (d, $J_{C,F}$ = 7.7 Hz), 64.3, 56.6; HR-MS (ESI): *m/z* [M+Na]$^+$ calcd for C$_{22}$H$_{16}$F$_3$IN$_2$NaO$_6$, 610.99028; found, 610.98869.

## Synthesis of optoA$_{2A}$RPAM-1 and optoA$_{2A}$RPAM-2 (Supplementary Fig. 5b)

**Bis(2-(trimethylsilyl)ethyl) 2,2'-((4-bromophenyl) azanediyl)diacetate (S3).** 2-(Trimethylsilyl)ethanol (6.35 mL, 44.5 mmol), HATU (20.2 g, 53.2 mmol), and *N,N*-diisopropylethylamine (18.5 mL, 107 mmol) were added to a stirred suspension of 2,2'-((4-bromophenyl)azanediyl)diacetic acid (**S2**, 5.07 g, 17.7 mmol) in dichloromethane (100 mL). The resulting mixture was stirred for 5 h at room temperature, diluted with water (50 mL), and extracted with chloroform. The combined organic layers were washed with brine, dried over anhydrous sodium sulfate, and concentrated in vacuo. The residue was subjected to column chromatography on silica gel (eluent: ethyl acetate/hexane, 10:90–20:80 v/v) to obtain bis(2-(trimethylsilyl)ethyl) 2,2'-((4-bromophenyl) azanediyl)diacetate (**S3**, 6.61 g, 76%) as a colorless oil; IR (NaCl): 2953, 2898, 1744, 1595, 1499, 1251, 1172, 859, 838 cm$^{-1}$; $^1$H NMR (400 MHz, CDCl$_3$) $\delta$ = 7.29 (d, *J* = 9.2 Hz, 2H), 6.48 (d, *J* = 9.2 Hz, 2H), 4.21–4.27 (m, 4H), 4.10 (s, 4H), 0.97–1.03 (m, 4H), 0.036 (s, 18H). $^{13}$C NMR (100 MHz, CDCl$_3$) $\delta$ = 170.7, 147.1, 132.0, 114.2, 110.5, 63.8, 53.8, 17.5, -1.44. HRMS (ESI): *m/z* [M+Na]$^+$ calcd for C$_{20}$H$_{34}$$^{79}$BrNNaO$_4$Si$_2$, 510.1107; found, 510.1107.

**Bis(2-(trimethylsilyl)ethyl) 2,2'-((4-(4,4,5,5-tetramethyl-1,3,2-dioxaborolan-2-yl)phenyl)azanediyl) diacetate (S4).** Bis(pinacolato) diboron (3.2 g, 12 mmol), potassium acetate (5.9 g, 59.7 mmol), and [1,1'-bis(diphenylphosphino)ferrocene]dichloropalladium(II) (PdCl$_2$(dppf), 435 mg, 594.7 µmol) were added to a stirred suspension of **S3** (5.8 g, 11.95 mmol) in 1,4-dioxane (150 mL). The resulting mixture was refluxed for 5 h and filtered through celite. The residue was subjected to column chromatography on silica gel (eluent: ethyl acetate/hexane, 5:95 to 15:85 v/v) to obtain bis(2-(trimethylsilyl)ethyl) 2,2'-((4-(4,4,5,5-tetramethyl-1,3,2-dioxaborolan-2-yl)phenyl)azanediyl) diacetate (**S4**, 4.1 g, 65%) as a yellow oil; IR (NaCl): 3436, 2956, 2901, 1739, 1724, 1618, 1588, 1526, 1410, 1357, 1249, 1216, 1172, 859, 837 cm$^{-1}$; $^1$H NMR (CDCl$_3$) $\delta$ = 7.67 (d, *J* = 8.7 Hz, 2H), 6.58 (d, *J* = 8.7 Hz, 2H), 4.20–4.27 (m, 4H), 4.13 (s, 4H), 1.31 (s, 12H), 0.96–1.04 (m, 4H), 0.032 (s, 18H). $^{13}$C NMR (CDCl$_3$) $\delta$ = 170.8, 150.2, 136.4, 111.5, 83.4, 63.7, 53.5, 24.9, 17.5, −1.44. HRMS (ESI): *m/z* [M+Na]$^+$ calcd for C$_{26}$H$_{46}$$^{11}$BNNaO$_6$Si$_2$, 558.2854; found, 558.2824.

**3-Bromo-7-(diethylamino)-4-(hydroxymethyl)-2H-chromen-2-one (S6).** A solution of 1 M tetra-*n*-butylammonium fluoride in tetrahydrofuran (23 mL), acetic acid (2.3 mL), and water (2.3 mL) was added to a stirred solution of 3-bromo-4-(((*tert*-butyldimethylsilyl)oxy)methyl)-7-(diethylamino)-2H-chromen-2-one[21] (**S5**, 6.67 g, 15.1 mmol) in tetrahydrofuran (207 mL). The mixture was stirred at room temperature for 19 h, diluted with a saturated aqueous solution of ammonium chloride (400 mL), and extracted with ethyl acetate. The combined organic layers were washed with brine, dried over anhydrous sodium sulfate, and concentrated in vacuo. The residue was subjected to column chromatography on silica gel (eluent: methanol/chloroform, 1:99 to 3:97 v/v) to give 3-bromo-7-(diethylamino)-4-(hydroxymethyl)-2H-chromen-2-one (S6, 3.99 g, 81%) as a yellow crystal; IR (KBr): 3434, 2967, 2908, 1702, 1685, 1618, 1578, 1523, 1433, 1407, 1357, 1268, 1225, 1159, 1072, 1042, 1014, 935, 819, 798, 757, 661, 549, 462 cm$^{-1}$; $^1$H NMR (CDCl$_3$) $\delta$ = 7.66 (d, *J* = 9.2 Hz, 1H), 6.61 (dd, *J* = 9.2, 2.8 Hz, 1H), 6.45 (d, *J* = 2.8 Hz, 1H), 4.94 (s, 2H), 3.40 (q, *J* = 7.1 Hz, 4H), 1.20 (t, *J* = 7.1 Hz, 6H); $^{13}$C NMR (CDCl$_3$) $\delta$ = 158.7, 155.3, 151.2, 150.9, 126.7, 109.5, 107.7, 105.3, 97.4, 61.8, 45.0, 12.6; HRMS (ESI): *m/z* [M+Na]$^+$ calcd for C$_{14}$H$_{16}$$^{79}$BrNNaO$_3$, 348.0211; found, 348.0215.

**Bis(2-(trimethylsilyl)ethyl) 2,2'-((4-(7-(diethylamino)-4-(hydroxymethyl)-2-oxo-2H-chromen-3-yl)phenyl)azanediyl)diacetate (A400-2).** Sodium carbonate (6.4 g, 75.8 mmol) and bis(triphenylphosphine)palladium(II) dichloride (PdCl$_2$(PPh$_3$)$_2$, 266 mg, 379 µmol) were added to a solution of **S4** (4.1 g, 7.6 mmol) and **S6**

(2.9 g, 8.9 mmol) in 1,4-dioxane/water (2:1 v/v, 300 mL). The resulting mixture was stirred for 1 h, cooled at room temperature, filtered through celite, and extracted with chloroform. The combined organic layers were washed with brine, dried over anhydrous sodium sulfate, and concentrated in vacuo. The residue was subjected to column chromatography on silica gel (eluent: ethyl acetate/hexane, 15:85 to 50:50 v/v) to obtain bis(2-(trimethylsilyl)ethyl) 2,2′-((4-(7-(diethylamino)-4-(hydroxymethyl)-2-oxo-2H-chromen-3-yl)phenyl)azanediyl) diacetate (A400-2, 2.9 g, 58%) as a yellow crystal; IR (KBr): 3470, 2960, 2892, 1744, 1705, 1611, 1582, 1523, 1410, 1352, 1278, 1165, 861, 831 cm$^{-1}$; $^1$H NMR (CDCl$_3$) δ = 7.67 (d, $J$ = 9.2 Hz, 1H), 7.21 (d, $J$ = 8.7 Hz, 2H), 6.61–6.67 (m, 3, H), 6.54 (d, $J$ = 2.8 Hz, 1H), 4.25 (d, $J$ = 5.7 Hz, 2H), 4.23–4.29 (m, 4H), 4.14 (s, 4H), 3.41 (q, $J$ = 7.1 Hz, 4H), 1.75, (t, $J$ = 5.7 Hz, 1H), 1.21 (t, $J$ = 7.1 Hz, 6H), 1.00–1.06 (m, 4H), 0.046 (s, 18H). $^{13}$C NMR (CDCl$_3$) δ = 171.1, 162.7, 155.7, 150.1, 147.7, 147.3, 131.4, 126.7, 123.5, 121.7, 112.2, 108.9, 108.0, 97.7, 63.7, 59.4, 53.6, 44.8, 17.5, 12.6, −1.43. HRMS (ESI): $m/z$ [M+Na]$^+$ calcd for C$_{34}$H$_{50}$N$_2$NaO$_7$Si$_2$, 677.3054; found, 677.3082.

**Bis(2-(trimethylsilyl)ethyl) 2,2′-((4-(7-(diethylamino)-4-(((3,4-difluoro-2-((2-fluoro-4-iodophenyl)amino)benzoyl)oxy)methyl)-2-oxo-2H-chromen-3-yl)phenyl)azanediyl)diacetate (optoA$_{2A}$RPAM-2).** N,N′-Dicyclohexylcarbodiimide (765 μL, 3.4 mmol) and 4-dinethylaminopyridine (278 mg, 2.3 mmol) were added to a solution of A400-2 (1.5 g, 2.8 mmol) and A$_{2A}$RPAM-1 (1.1 g, 2.7 mmol) in dichloromethane (50 mL) in the dark. The resulting mixture was stirred for 24 h at room temperature, diluted with water (50 mL), and extracted with chloroform. The combined organic layers were washed with brine, dried over anhydrous sodium sulfate, and concentrated in vacuo. The residue was subjected to column chromatography on silica gel (ethyl acetate/hexane, 10/90 to 50/50 v/v) to obtain bis(2-(tri-methylsilyl)ethyl) 2,2′-((4-(7-(diethylamino)-4-(((3,4-difluoro-2-((2-fluoro-4-iodophenyl)amino)benzoyl)oxy)methyl)-2-oxo-2H-chromen-3-yl)phenyl)azanediyl)diacetate (**optoA$_{2A}$RPAM-2**, 1.3 g, 58%) as a yellow crystal; IR (KBr): 3314, 2954, 2896, 1742, 1719, 1619, 1597, 1525, 1500, 1409, 1358, 1271, 1186, 1163, 1146, 1066, 982, 859, 837, 775 cm$^{-1}$; $^1$H NMR (CDCl$_3$) δ = 8.92 (br, 1H), 7.70 (ddd, $J$ = 9.2, 6.0, 1.9 Hz, 1H), 7.40 (dd, $J$ = 10.1, 1.8 Hz, 1H), 7.35 (d, $J$ = 8.7 Hz, 2H), 7.22 (d, $J$ = 8.7 Hz, 2H), 6.66–6.77 (m, 2H), 6.62 (d, $J$ = 8.7 Hz, 2H), 6.52–6.57 (m, 2H), 4.25 (s, 2H), 4.21–4.28 (m, 4H), 4.12 (s, 4H), 3.41 (q, $J$ = 7.1 Hz, 4H), 1.20 (t, $J$ = 7.1 Hz, 6H), 0.98–1.04 (m, 4H), 0.030 (s, 18H); $^{13}$C NMR (CDCl$_3$) δ = 171.0, 166.3 (d, $J_{C,F}$ = 1.9 Hz), 162.2, 155.4, 154.6 (dd, $J_{C,F}$ = 255.9, 11.5 Hz), 154.3 (d, $J_{C,F}$ = 250.2 Hz), 149.9 (br), 148.1, 142.1, 142.0 (dd, $J_{C,F}$ = 250.2, 14.4 Hz), 136.2 (dd, $J_{C,F}$ = 7.7, 3.8 Hz), 133.0 (d, $J_{C,F}$ = 2.9 Hz), 131.5, 129.9 (d, $J_{C,F}$ = 8.6 Hz), 127.8 (dd, $J_{C,F}$ = 9.6, 3.8 Hz), 126.0, 124.6 (d, $J_{C,F}$ = 22.0 Hz), 124.3 (br), 122.9, 122.1 (d, $J_{C,F}$ = 5.8 Hz), 113.4, 112.2, 109.5 (br), 108.6 (d, $J_{C,F}$ = 18.2 Hz), 98.1 (br), 84.0 (d, $J_{C,F}$ = 7.7 Hz), 63.8, 61.7, 53.6, 45.3, 17.6, 12.5, −1.4; HRMS (ESI): $m/z$ [M+Na]$^+$ calcd for C$_{47}$H$_{55}$F$_3$IN$_3$NaO$_8$Si$_2$, 1052.2422; found, 1052.2428.

**2,2′-((4-(7-(Diethylamino)-4-(((3,4-difluoro-2-((2-fluoro-4-iodophenyl)amino)benzoyl)oxy)methyl)-2-oxo-2H-chromen-3-yl)phenyl)azanediyl)diacetic acid (optoA$_{2A}$RPAM-1).** Zinc (II) chloride (159 mg, 1.16 mmol) was added to a solution of optoA$_{2A}$RPAM-2 (100 mg, 90 μmol) in 2,2,2-trifluoroethanol (10 mL) in the dark. The resulting mixture was stirred for 24 h at room temperature. The mixture was diluted with a saturated aqueous solution of ammonium chloride (10 mL) and extracted with chloroform. The organic layers were combined, washed with brine, dried over anhydrous sodium sulfate, and concentrated in vacuo. The residue was subjected to column chromatography on reverse-phase silica gel (eluent: water/acetonitrile, 100:0 to 50:50 v/v) to obtain 2,2′-((4-(7-(diethylamino)-4-(((3,4-difluoro-2-((2-fluoro-4-iodophenyl)amino)benzoyl)oxy)methyl)-2-oxo-2H-chromen-3-yl)phenyl)azanediyl)diacetic acid (**optoA$_{2A}$RPAM-1**, 55.3 mg, 69%) as a yellow solid; IR (KBr): 3403 (br), 2969, 2926, 1696, 1663, 1618, 1589, 1526, 1502, 1270, 1191, 1138, 1068, 975, 819, 776, 663, 630, 610, 545, 439 cm$^{-1}$; $^1$H NMR (CD$_3$OD) δ = 8.58 (s, 1H, NH), 7.74 (ddd, $J$ = 9.2, 6.0, 2.3 Hz, 1H), 7.47 (d, $J$ = 9.2 Hz, 1H), 7.38 (dd, $J$ = 10.5, 1.8 Hz, 1H), 7.32 (d, $J$ = 8.2 Hz, 1H), 7.21 (d, $J$ = 8.7 Hz, 2H), 6.95 (ddd, $J$ = 9.2, 9.2, 7.3 Hz, 1H), 6.59–6.69 (m, 4H), 6.56 (d, $J$ = 2.3 Hz, 1H), 5.31 (s, 2H), 4.18 (s, 4H), 3.47 (q, $J$ = 7.0 Hz, 4H), 1.21 (t, $J$ = 7.0 Hz, 6H); $^{13}$C NMR (DMSO-$d6$) δ = 174.3, 165.0, 161.1, 154.8, 153.2 (dd, $J_{C,F}$ = 252.1, 11.5 Hz), 152.7 (d, $J_{C,F}$ = 247.3 Hz), 149,8, 146.6, 143.1 (dd, $J_{C,F}$ = 248.7, 13.4 Hz), 133.5 (dd, $J_{C,F}$ = 6.7, 2.9 Hz), 133.1 (d, $J_{C,F}$ = 2.9 Hz), 131.4, 130.8 (d, $J_{C,F}$ = 10.5 Hz), 127.7 (d, $J_{C,F}$ = 6.7 Hz), 126.6, 123.6 (d, $J_{C,F}$ = 21.1 Hz), 122.5, 121.2, 120.7, 117.7, 110.6 (d, $J_{C,F}$ = 18.2 Hz), 110.2, 108.9, 107.0, 96.6, 82.8 (d, $J_{C,F}$ = 7.7 Hz), 61.5, 57.1, 44.0, 12.4; HRMS (ESI): $m/z$ [M-H]$^-$ calcd for C$_{37}$H$_{30}$F$_3$IN$_3$O$_8$, 828.1030; found, 828.1059.

**Formation of the sodium salt of optoA$_{2A}$RPAM-1.** Sodium carbonate in a 100-mM aqueous solution (835 μL, 83.5 μmol) was added to a solution of optoA$_{2A}$RPAM-1 (69.3 mg, 83.5 μmol) in tetrahydrofuran (42 mL) in the dark. The resulting mixture was stirred for 30 min at 0 ˚C. Diethylether (30 mL) was then added to the mixture. The precipitate was filtered and freeze-dried to obtain the sodium salt of optoA$_{2A}$RPAM-1 (56.8 mg, 67%) as a yellow amorphous solid. Elemental Anal. Calcd for C$_{37}$H$_{29}$F$_3$IN$_3$Na$_2$O$_8$·7.5H$_2$O: C, 48.06; H, 4.40; N, 4.17. Found: C, 43.89; H, 4.78; N, 3.96.

### Photo-uncaging reaction

All experiments were performed in the dark or under light-shielded conditions. The uncaging reaction of an aqueous solution of Nv-A$_{2A}$RPAM-1 or optoA$_{2A}$RPAM-1 (200 μM, 3 mL) was analyzed in a quartz cell (3.5 mL, 1-cm light path) under stirring and irradiation with 365-nm light (SLUV-8, AS ONE Co.) or 405-nm light (UHP-M-405, Prizmatix), respectively. The solution was combined with 3-isobutyl-1-methylxanthine (IBMX) in DMSO (10 mM, 20 μL) as an internal standard and filtered through a Millex syringe filter (PTFE, 0.45 μm pore size, 4 mm diameter, Millipore). The mixture was injected into a Waters ACQUITY UPLC system with photodiode array detection at 254 nm. An ACQUITY UPLC BEH C18 column (1.7 μm, 50 mm × 2.1 mm; Waters) maintained at 40 °C with a graded aqueous 0.05% (w/w) trifluoroacetic acid/acetonitrile mobile phase [Mobile phase A: 10% acetonitrile in water (v/v) + 0.05% (w/w) trifluoroacetic acid, mobile phase B: acetonitrile, gradient: 70% B to 100% B in 5 min] was used for UPLC separation. Calibration curves were created from the peak area ratio of optocompound or A$_{2A}$RPAM-1 to that of the internal standard to determine the concentration of optocompound and A$_{2A}$RPAM-1 in each sample.

### Actinometry

All experiments were performed in the dark or under light-shielded conditions. To determine the light intensity ($I$) of the UHP-M-405-nm light source (Prizmatix), potassium trioxalato ferrate (148 mg, 302 μmol) was weighed into a 50-mL volumetric flask and aqueous sulfuric acid (50 mM) was added to make exactly 50 mL (solution A). 1,10-Phenanthroline (50 mg) and sodium acetate (2.04 g) were weighed into 50 mL vial and dissolved in water (24.7 mL) and aqueous sulfuric acid (180 mM, 0.24 mL) (solution B). Solution A (3 mL) was irradiated in a quartz cell (3.5 mL, 1-cm light path) with 405-nm light (UHP-M-405 with a 1% attenuation filter) for 0, 1, 2, and 3 s, and 0.5 mL of solution B was subsequently added. The photochemical formation of iron (II) was determined by measuring the characteristic absorption of the iron/1,10-phenanthroline complex at 510 nm as a function of Fe$^{2+}$ concentration.

### Quantum yield analysis

To determine the quantum yield ($\phi$), a solution of optoA$_{2A}$RPAM-1 (312 μM, 3 mL) was irradiated with 405-nm light (UHP-M-405 with a 1% attenuation filter, $I$ = 1.57 × 10$^{-6}$ mol/s) and analyzed by UPLC. The

number of optoA$_{2A}$RPAM-1 molecules that reacted was determined by UPLC and the quantum yield was calculated according to the following formula:

$$\phi = \frac{N}{I \times T}$$

$N$: number of molecules reacted (mol), $I$: light intensity (mol/s), $T$: irradiation time (s)

## Animals

Male C57BL/6, A$_{2A}$R KO[53], vesicular γ-aminobutyric acid transporter-Cre[54], and MAP6 KO[10] mice (10–15 weeks of age, 24–30 g) used in the experiments were maintained at the International Institute for Integrative Sleep Medicine and housed in insulated sound-proof chambers maintained at an ambient temperature of $23 \pm 0.5$ °C with $50 \pm 5\%$ humidity on a 12-h light/dark cycle (lights on at 8:00, illumination intensity ≈ 100 lux). Food and water were provided *ad libitum*. All experiments were performed in accordance with the Animal Care Committee of the University of Tsukuba (study protocol #23–268) and the US National Institutes of Health Guidelines for the Care and Use of Laboratory Animals[55]. Every effort was made to minimize the number of animals used as well as any pain or discomfort.

## cAMP assay

Activation of A$_{2A}$Rs was quantified based on cAMP accumulation in CHO cells expressing mouse A$_{2A}$Rs generated in a previous study[9]. CHO cells were suspended in Hank's balanced salt solution containing 1 M HEPES and 0.25 M 3-isobutyl-1-methylxanthine in 384-well microplates ($2 \times 10^3$ cells/well) and incubated with adenosine and A$_{2A}$RPAM-1 or optoA$_{2A}$RPAM-1 after light irradiation for the indicated time for 30 min at 25 °C. After adding the detection mixture containing the Eu-cAMP tracer and ULight-anti-cAMP antibody, the plates were further incubated for 1 h at 25°C. A microplate reader (ARVO X5, PerkinElmer; excitation: 340 nm; emission: 665 nm) was used to measure the Förster resonance energy transfer signal. All experiments were performed according to the manufacturer's instructions (LANCE Ultra cAMP Kit, PerkinElmer). The cAMP levels are based on the dynamic range ("linear portion") of the cAMP standard curve and normalized to the baseline or adenosine-treated group.

## Patch-clamp electrophysiology

C57BL/6 mice were deeply anesthetized with enflurane, and their brains were promptly removed and placed in ice-cold aCSF, composed of 124 mM sodium chloride, 26 mM sodium hydrogen carbonate, 3 mM potassium chloride, 2 mM calcium chloride, 1 mM magnesium sulfate, 1.25 mM potassium dihydrogen phosphate, and 10 mM glucose; osmolarity was adjusted to 300 to 310 mOsm; and equilibrated with 95% oxygen and 5% carbondioxide. The brains were mounted on the stage of a vibrating microtome (Leica VT1200 S) with cyanoacrylate. Coronal sections containing the NAc and striatum (250 μm thick) were obtained and allowed to recover in aCSF at room temperature. Slices were submerged in a perfusion chamber placed under an upright microscope (BX51WI; Olympus) fitted with a custom LED IR illumination and Nomarski interference contrast. Slices were superfused with aCSF at a rate of 1 mL min$^{-1}$. Patch electrodes had a resistance of 4−5 MΩ with an internal solution containing 130 mM potassium gluconate, 1 mM egtazic acid, 10 mM HEPES, 5 mM magnesium ATP, 0.5 mM sodium GTP, and 5 mM sodium chloride. The pH was adjusted to 7.3 with sodium chloride and osmolarity to 290−300 mOsm. Signals were amplified (Axopatch 700 A; Molecular Devices) and then digitized using customized routines in commercial software (IGOR Pro; WaveMetrics). Signals were filtered at 10 kHz and digitized at 20 kHz. A$_{2A}$RPAM-1 or optoA$_{2A}$RPAM-1 were added to the aCSF bath and irradiated with 405-nm light (3-s pulses delivered every 60 s) delivered via an optical fiber coupled to a LED source (UHP-Mic-LED-405, Prizmatix) and coupled to an epifluorescence attachment of the microscope. A$_{2A}$R$^+$ and A$_{2A}$R$^−$ neurons were distinguished by their electrophysiologic characteristics[56].

## Surgery

The mice used in the behavioral experiments were anesthetized with isoflurane (4% for induction, 2% for maintenance) and placed in a stereotaxic apparatus for implantation procedures. For the focal injection of adenosine, adenosine receptor agonists, the A$_{2A}$R antagonist, and fluorescein, double stainless steel guide cannulas (62032, RWD Life Science) were stereotaxically implanted into the NAc with the following coordinates: 1.5 mm anterior and to the bregma and 4.1 mm below the dura. Dummy cannulas (62132, RWD Life Science) were inserted into the guide cannulas after surgery and removed only during microinjections. Microinjections of vehicle or drugs were performed using a tubing-nested Hamilton 10-μL syringe connected to an internal cannula (62232, RWD Life Science), and the solutions were slowly injected manually into freely behaving mice.

Mice were stereotaxically injected bilaterally into the NAc 1.5 mm anterior and 1.2 mm lateral to the bregma and 4.1 mm below the dura[57] with recombinant AAVDj8-GFAP-hM3Dq-mCherry (264 nl/side, $9.1 \times 10^{10}$ particles ml$^{-1}$), AAV10-hsyn-hM3Dq-mCherry (264 nl/side, $4.1 \times 10^{11}$ particles ml$^{-1}$), AAV10-hsyn-hM4Di-mCherry (264 nl/side, $1.1 \times 10^{11}$ particles ml$^{-1}$), AAVDj8-GFAP-ChR2-mCherry (264 nl/side, $6.5 \times 10^{10}$ particles ml$^{-1}$), AAV9-hSyn-GRAB$_{Ado1.0}$, (200 nl/side, $5.1 \times 10^{10}$ particles ml$^{-1}$, 1:1 mixed with GFAP-hM3Dq, hSyn-hM3Dq, hSyn-hM4Di or GfaABC1D-hPMCA2w/b AAVs for adenosine fiber photometry), AAV9-hSyn-GRAB$_{Ado1.0mut}$ (200 nl/side, $4.3 \times 10^{10}$ particles ml$^{-1}$, 1:1 mixed with GFAP-hM3Dq for adenosine fiber photometry), AAV2/9-GFAP-hPMCA2w/b-mCherry (200 nl/side, $5 \times 10^{12}$ particles ml$^{-1}$) or AAV10-hSyn-DIO-hM3Dq-mCherry (264 nl/side, $2.4 \times 10^{10}$ particles ml$^{-1}$), using a glass micropipette and an air pressure injector system[12,58].

To selectively target the NAc with the optodialysis probe, we unilaterally implanted a CMA7 microdialysis guide cannula (Harvard Bioscience) attached with an implantable fiber optic cannula (core diameter 200 μm) above the NAc 1.2 mm anterior and 1.2 mm lateral to the bregma and 3.5 mm below the dura[57] so that the tip of the optical fiber protruded 0.5 mm from the end of the cannula during light stimulation[13]. At the time of the behavioral experiment, the dummy cannula was replaced by a CMA7 6-kDa microdialysis probe with a 1-mm-long dialysis membrane (Harvard Biosciences) extending below the guide cannula. For other optochemical experiments, 2 fiber optic cannulas (core diameter 200 μm; Kyocera) were bilaterally placed during surgery above the NAc, located 1.2 mm anterior and 1.2 mm lateral to bregma and 4 mm below the dura.

For monitoring sleep/wake behavior, mice were chronically implanted with EEG and EMG electrodes. Briefly, the implant comprised 2 stainless-steel screws (1 mm diameter) serving as EEG electrodes, 1 placed epidurally over the right frontal cortex (1 mm anterior and 1.6 mm lateral to bregma) and the other placed over the right parietal cortex (1 mm anterior and 1.6 mm lateral to lambda). Two insulated Teflon-coated, silver wires (0.2 mm in diameter), placed bilaterally into the trapezius muscles, served as EMG electrodes. EEG and EMG electrodes were connected to a microconnector, and the assembly was then fixed to the skull with self-curing dental acrylic resin. The optodialysis probe did not interfere with the implantation of the EEG electrodes and allowed the mouse to move freely in the cage.

## Microdialysis

Microdialysis was performed as previously described[29]. Briefly, under pentobarbital anesthesia (60 mg kg$^{-1}$, intraperitoneal), we implanted bilaterally guide cannulas (0.40 mm inner diameter, 0.50 mm outer diameter; Eicom) above the NAc 1.4 mm anterior and 1.2 mm lateral

to the bregma and 2.9 mm below the dura[57]. At the time of the behavioral experiment, the mouse was quickly anesthetized using isoflurane and the dummy cannula was removed followed by insertion of the microdialysis probe (1 mm membrane length; Eicom) into the guide cannula. The probe was infused continuously with Ringer's solution using an infusion pump at a flow rate of 0.5 μl min$^{-1}$. Two hours after inserting the probe, dialysates were continuously collected from the probe for 3 h. The dialysates were kept at −20 °C until the UPLC analysis was performed.

A Shimadzu UPLC system equipped with a UV detection system and a TSKgel ODS-100V UPLC column (Tosoh Bioscience) and maintained with an aqueous acetonitrile mobile phase containing 100 mM monosodium phosphate (aqueous to organic solvent ratio 96:4) at a flow rate of 1 ml min$^{-1}$ was used for the UPLC separation. 80 μL of each dialysate or adenosine standard was injected into the UPLC system and calibration curves were constructed from the peak area ratio of adenosine to determine the adenosine concentration in each sample.

## Behavioral experiments

Adenosine (Merck), CGS 21680 [(3-(4-(2-((6-Amino-9-(N-ethyl-β-D-ribofuranosyluronamide)-9H-purin-2-yl)amino)ethyl)phenyl)propanoic acid; Merck], CPA (Merck), and ZM 241385 [4-(2-{[7-Amino-2-(furan-2-yl)(1,2,4)triazolo(1,5-a)(1,3,5)triazin-5-yl]amino}ethyl)phenol; Abcam] were dissolved in aCSF containing 5% DMSO, CNO (Merck) and fluorescein (Merck) were dissolved in saline, the optoA$_{2A}$RPAM-1 was dissolved in Ringer's solution (optodialysis) or saline (intraperitoneal injection), and optoA$_{2A}$RPAM-2 was dissolved in a 1% aqueous solution of Soluplus containing 5% DMSO, 5% Cremophor, and 20% polyethylene glycol 400. All experiments with caged compounds were performed under red light to avoid photolysis of the compounds. Specific drug doses used in each experiment are mentioned in the results section and Figure legends.

EEG/EMG recordings were performed according to a previous study[59]. Briefly, after allowing 1–2 weeks for postoperative recovery, the mice were connected with EEG/EMG recording cables. The EEG/EMG signals were amplified and filtered by an amplifier (Biotex; EEG: 0.5–64 Hz, EMG: 16–64 Hz), digitized at a sampling rate of 128 Hz, and recorded using SLEEPSIGN software (Kissei Comtec). Vigilance states were scored offline by characterizing 10-s epochs into 3 stages: wake, SWS, and REMS, according to standard criteria[59].

For DREADD experiments, all AAV-injected mice were injected intraperitoneally with vehicle or CNO at 19:00 on 2 consecutive days. On day 1, mice were treated with saline and on day 2, mice were treated with CNO.

For opto-experiments, violet light (405 nm) was generated by a UHP-Mic-LED-405 light source (Prizmatix) and applied through optical fibers (Lucir, 250 μm diameter). A pulse generator (Grass-Natus Neurology Incorporated) was used to control the duration and frequency of light pulses. Fiber-optic rotary joints (Lucir) were used for unrestricted in-vivo illumination. Violet light power intensity at the tip of the plastic fiber was ~15 mW mm$^{-2}$, measured by a power meter. Specific light frequencies and duration used in each experiment are mentioned in the Results section and Figures.

For optodialysis, the probe was continuously infused using an infusion pump with Ringer's solution[60] for 24 h (8:00–8:00) at a speed of 2 μl min$^{-1}$. OptoA$_{2A}$RPAM-1 (10 nmol min$^{-1}$) was infused into the NAc for 1.5 h (21:00–22:30), and light was applied for 1 h (21:30–22:30). Recordings made on a different day during the infusion of Ringer's solution alone served as the control for the same animal. The optodialysis probe remained in the guide cannula on non-experimental days and was infused with Ringer's solution at a speed of 0.1 μl min$^{-1}$.

## Fiber photometry recording and analysis

For at least 1 week after EEG/EMG implantation and AAV injection, the mice were housed individually in transparent cages in a sound-proof recording chamber. The mice were acclimated to the recording cable for at least 3 days before starting the recording process. For spontaneous and astrocytic activation-evoked adenosine release, the cortical EEG and EMG signals were recorded as described for the behavioral experiments. To record fluorescence from the adenosine sensor, we attached an optical fiber (FT200UMT, Thorlabs or OFJ-F-B-200-1.25-0.37.1 C.200, Hangzhou Sanshi Biotechnology Co.) to the implanted ferrule through a ceramic sleeve and recorded the emission fluorescence at 50 Hz using a commercial fiber photometry system (Thinker Tech Nanjing Biotech Co.) or a Power 1401 digitizer and Spike2 software (Cambridge Electronic Design). Fluorescence recordings were performed on freely moving mice to measure spontaneous and chemogenetic activation-evoked adenosine release. The signal from each continuous experimental trial was normalized to the average fluorescence using a MATLAB program [Supplementary software is available in the Figshare database (https://doi.org/10.6084/m9.figshare.25468084)][61]. Briefly, the raw signals were first adjusted to account for photo-bleaching by considering the overall trend before further analysis. We obtained fluorescence change values (ΔF/F) by calculating (F − F0)/F0, where the baseline fluorescence signal (F0) was the average signal obtained during SWS sleep. Peri-event plots were generated to display ΔF/F values.

## Open field test

Mice were individually introduced to a novel open field environment (dimensions: 44 × 44 × 30 cm) located in a dimly lit room, with a 10-min exploration period. A digital video camera positioned above the arena was connected to a computer running video tracking software (SMART v3.0.02, Panlab). This setup facilitated the monitoring of the mouse's movements and allowed the determination of time and distance traveled in 2 defined areas: the periphery (within 15 cm from the walls) and the central zone (14 cm × 14 cm at the center of the open field). To ensure cleanliness and eliminate olfactory cues, the open field arena was carefully wiped with 70% ethanol and dried between each mouse.

## UPLC-MS/MS analysis

The brains of mice injected intraperitoneally with A$_{2A}$RPAM-1, optoA$_{2A}$RPAM-1 or optoA$_{2A}$RPAM-2 at 21:00 were collected 30 min after treatment. Each brain was combined with 300 μL of acetonitrile with 1% formic acid (v/v) and vortexed for 3 min under normal light conditions to convert all brain-penetrating optoA$_{2A}$RPAM-1 or optoA$_{2A}$RPAM-2 to A$_{2A}$RPAM-1. The precipitated proteins were then removed by centrifugation (1000 × $g$ for 5 min), and the supernatant was transferred to a HybridSPE-Phospholipid Ultra cartridge (Supelco) and eluted from the cartridge by applying a vacuum. The eluate was injected into a Waters ACQUITY UPLC-MS/MS system with an electrospray ionization interface and operated in the negative ion mode. An ACQUITY UPLC BEH C18 column (1.7 μm, 50 mm × 2.1 mm; Waters) with a graded acetonitrile/water mobile phase at a flow rate of 500 μL min$^{-1}$ was used for UPLC separation. A$_{2A}$RPAM-1 was detected by single ion (m/z 392) monitoring.

## Histology

For histologic analyses, the mice were deeply anesthetized with an overdose of chloral hydrate (500 mg kg$^{-1}$, intraperitoneal injection) and perfused through the left ventricle of the heart with saline, followed by neutral buffered 10% formalin. The brains were removed and placed in 20% sucrose in phosphate-buffered saline overnight at 4 °C to reduce freezing artifacts. The brains were then sectioned at 40 μm on a freezing microtome. Immunohistochemistry was performed on free-floating sections as described previously[29]. In brief, the sections were rinsed in 0.25% Triton X-100 in PBS (PBT) and incubated in PBT containing 10% BlockAce (DS PharmaBiomedical) for 30 min at room temperature. The sections were then incubated with the rabbit anti-GFAP antibody (1:200, Cat#HPA056030, Merck), goat anti-mCherry antibody (1:1000, Cat#

AB0040-200, SICGEN), mouse anti-NeuN antibody (1:100, Cat# MAB377, Millipore) containing 5% BlockAce at room temperature in the combinations described in the Results section. After overnight incubation, the sections were rinsed in PBT and incubated with donkey anti-goat Alexa Fluor 594 nm (1:1000, Cat# A11058, Thermo Fisher Scientific), donkey anti-mouse Alexa Fluor® 647 nm (1:500, Cat# A31571, Thermo Fisher Scientific), or donkey anti-rabbit Alexa Fluor® 647 nm (1:500, Cat# A31573, Thermo Fisher Scientific) containing 5% BlockAce for at least 2 h. The sections were then mounted on glass slides and sealed with mounting medium containing DAPI dye (Vector Labs) and cover glass. Fluorescence signals were visualized using an LSM 700 confocal microscope (Zeiss, Oberkochen, Germany).

For histologic verification of the optodialysis probe placement, the sections were mounted on glass slides, stained with 0.1% cresyl violet solution (Merck), and differentiated with 10% acetic acid in ethanol. The brain sections were finally dehydrated using graded ethanol, fixed in xylene, and cover-slipped using malinol (Muto Pure Chemicals).

### Statistical analysis
Statistical analysis was performed using Graph Pad Prism 10.2.1 (Dotmatics). All data were subjected to the Kolmogorov-Smirnov test for Gaussian distribution and variance. Comparisons between 2 groups were performed using the unpaired 2-tailed Student's t-test, 2-tailed Mann-Whitney U test, and Wilcoxon signed-rank test. Comparisons among multiple parameters were performed by a 2-way repeated-measures analysis of variance (ANOVA) followed by Bonferroni's, Tukey's, and Benjamini-Hochberg's *post hoc comparisons*. All measurements were taken from distinct samples.

### Reporting summary
Further information on research design is available in the Nature Portfolio Reporting Summary linked to this article.

## Data availability
The source data generated in this study have been deposited in the Figshare database [https://doi.org/10.6084/m9.figshare.25468084][61]. Additional data that support and extend the findings of this study, such as source spectrometric data for the synthesized chemical compounds and polysomnographic recordings, are available from the corresponding authors upon request.

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

## Acknowledgements

We thank all lab members and members of the strategic basic research project "[OptBio] Development and application of optical technology for spatiotemporal control of biological functions" for their technical assistance, discussions, and comments. We thank Sara Kobayashi for helping to prepare the Figures and Dong Hui for providing MATLAB code. This work was supported by the Japan Science and Technology Agency (CREST [grant numbers JPMJCR1655 to M.Y. and M.L. and JPMJCR18R4 to M.Ab.); the Natural Science Foundation of China (grant numbers 82101556 to X.Z. and 82171479 and 82020108014 to Y.W.); the Japan Society for the Promotion of Science (KAKENHI [grant numbers JP21H02802 to M.L., JP23K14685 to K.R., JP18K14352 to T.S. and JP17H03022, JP20K21197, JP21H01921, and JP22K19033 to M.Ab.], Grant-in-Aid for Transformative Research Areas [Glia decoding: deciphering information critical for brain-body interaction, grant number JP23H04148] to M.L., and RECONNECT Initiative [grant number JP22K21351] to M.Y., T.S., and M.L.); the Japan Agency for Medical Research and Development (AMED) Moonshot Program (grant number JP21zf0127005) and Strategic Promotion Program for Translational Research (grant number JP24ym0126803) to M.L.; the Chan Zuckerberg Initiative under the Collaborative Pairs Pilot Project Awards (Cycle 2) to M.L.; the Brain Science Foundation (to K.R.); the Organization for the Promotion of Strategic Research Initiatives of the University of Tsukuba (to K.R.); the project "Social Application of Mobility Innovation and Future Social Engineering Research Phase IV" (grant number CRIO4006), a joint research project between Toyota Motor Corporation and the University of Tsukuba to M.L.; the Japan Foundation for Applied Enzymology (grant number 16H007) to T.S.; the World Premier International Research Center Initiative (WPI) from the Ministry of Education, Culture, Sports, Science and Technology (MEXT) to M.Y., T.S., and M.L.; the Young Runners in Strategy of Transborder Advanced Researches (TRiSTAR) program from MEXT to T.S.; the Natural Science Foundation of Zhejiang Province of China (grant number LQ22H090013) to X.Z.; the STI2030-Major Project (grant number 2021ZD0203400) to Y.W.; the National Key Research and Development Program of China (grant number 2022YFA1604504) to Y.W.; the Lingang Laboratory & National Key Laboratory of Human Factors Engineering Joint Grant (grant number LG-TKN-202203-01) to Y.W.; the National Key R&D Program of China (grant number 2022YFE0108700) to Y.L.; the Uehara Memorial Foundation to M.Ab.; the United States Department of Veterans Affairs (Merit

awards [grant numbers I01BX001404 and I01BX0061950] to R.B. and Carrier Development Award [grant number IK2 BX004905] to D.S.U.); and the National Institutes of Health (grant number R01 NS119227) to R.B.

## Author contributions

K.R., X.Z., T.S., Y.W., M.Ab., R.B., and M.L. designed the experiments. K.R., X.Z., P.Y., R.O., S.I., T.K., N.H.T.F., H.I., M.Am., D.S.U., and K.V. performed the experiments and analyzed the data. Z.W., Y.L., Y.Che., R.B., and Y.Chi. provided the reagents. M.Ab., H.N., and M.Y. provided critical feedback and helped shape the research, analysis, and manuscript. K.R., Y.W., T.S., and M.L. wrote the manuscript. All authors approved the final version of the manuscript to be published.

## Competing interests

The authors declare no competing interests.

## Additional information

[1]International Institute for Integrative Sleep Medicine (WPI-IIIS), University of Tsukuba, Tsukuba, Ibaraki, Japan. [2]Oujiang Laboratory (Zhejiang Laboratory for Regenerative Medicine, Vision and Brain Health), School of Ophthalmology & Optometry and Eye Hospital, Wenzhou Medical University, Wenzhou, Zhejiang, China. [3]Department of Pharmacology, School of Basic Medical Sciences, State Key Laboratory of Medical Neurobiology, Institutes of Brain Science and Collaborative Innovation Center for Brain Science, Joint International Research Laboratory of Sleep, Fudan University, Shanghai, China. [4]School of Pharmacy, Wannan Medical College, Wuhu, China. [5]PhD Program in Humanics, University of Tsukuba, Tsukuba, Ibaraki, Japan. [6]State Key Laboratory of Molecular Developmental Biology, Institute of Genetics and Developmental Biology, Chinese Academy of Sciences, Beijing, China. [7]Department of Chemistry, Graduate School of Advanced Science and Engineering, Hiroshima University Research Center for Photo-Drug-Delivery Systems (HiU-P-DDS), Hiroshima University, Higashi-Hiroshima, Hiroshima, Japan. [8]Department of Psychiatry, Veterans Administration Boston Healthcare System and Harvard Medical School, West Roxbury, MA, USA. [9]New Cornerstone Science Laboratory, State Key Laboratory of Membrane Biology, School of Life Sciences, Peking University, Beijing, China. [10]Institute of Medicine, University of Tsukuba, Tsukuba, Ibaraki, Japan. [11]These authors contributed equally: Koustav Roy, Xuzhao Zhou, Rintaro Otani, Ping-Chuan Yuan. ✉e-mail: yiqunwang@fudan.edu.cn; saito.tsuyoshi.gf@u.tsukuba.ac.jp; lazarus.michael.ka@u.tsukuba.ac.jp

