## [Peer Review File · Nature Communications]

Optochemical control of slow-wave sleep in the nucleus accumbens of male mice by a photoactivatable allosteric modulator of adenosine A2A receptorsREVIEWER COMMENTS

Reviewer #1 (Remarks to the Author):

The team in their previous work (Korkutata et al., *Neuropharmacology*, 2019) identified a positive allosteric modulator of A2ARs that can penetrate the blood-brain barrier and named it A2AR PAM-1. They found that both intracerebroventricular and intraperitoneal administration of A2AR PAM-1 increased slow-wave sleep (SWS) in mice, and the sleep-promoting effect of A2AR PAM-1 depended on A2AR. In this manuscript, the authors synthesized two derivatives of A2AR PAM-1, named optoA2ARPAM-1 and optoA2ARPAM-2, which can be "activated" by visible light (~400nm). They validated the effectiveness of these compounds in regulating sleep-wake behavior in nucleus accumbens (NAc) in freely moving animals, providing informative data for the potential application of photopharmacology in basic research and clinical settings. The main findings of this manuscript are as follows:

1. Extracellular adenosine signaling in the NAc exerts a sleep-promoting effect, dependent on adenosine A2A receptors rather than A1 receptors.
2. Both astrocytic and neuronal activity in the NAc contribute to the increase in extracellular adenosine levels.
3. Local infusion of optoA2ARPAM-1 in the NAc combined with light stimulation led to an increase in SWS in mice.
4. Intraperitoneal injection of optoA2ARPAM-2 combined with light stimulation in the NAc resulted in increased SWS in mice, and this process depended on A2A receptors.

Overall, this manuscript is quite interesting, but I have the following questions:

1. In Fig.1, I noticed that the adenosine concentration infused in the NAc was 3.5mM. However, if I'm not mistaken, the extracellular adenosine concentration under physiological conditions is only in the range of a few hundred nanomolar, and even in some pathological conditions, it can only reach the micromolar level. Therefore, is the increase in SWS induced by adenosine infusion comparable to SWS under physiological conditions (electroencephalogram)? Furthermore, how does the temporal profile of slow-wave sleep (SWS) change in mice after intracerebroventricular infusion of adenosine or adenosine receptor agonists in NAc? Please provide a graph similar to the one shown in Figure 1h.
2. Based on some gain-of-function experiments, the authors concluded the following from Figure 1: 1) Adenosine exerts a sleep-promoting effect in the NAc, dependent on A2A receptors rather than A1 receptors; 2) Both astrocytic and neuronal activity in the NAc contribute to the increase in extracellular adenosine levels. However, these conclusions lack loss-of-function experiments to validate them. For example, what is the effect of adenosine receptor antagonists infused in the NAc on mouse sleep? Which class of adenosine receptors is responsible for blocking the increase in slow-wave sleep (SWS) induced by adenosine infusion? How do inhibiting astrocytic and neuronal activity in the NAc affect extracellular adenosine concentration and mouse sleep?
3. Did the authors investigate the role of A1R in the chemogenetic activation-induced sleep increase in NAc astrocytes?
4. Since NAc adenosine has a sleep-promoting effect, I'm curious about the changes in NAc adenosine concentration during the normal sleep-wake cycle.
5. In Fig.1, The authors employed chemogenetic techniques to activate astrocytes in the NAc and simultaneously utilized microdialysis to monitor changes in extracellular adenosine concentrations in the same brain region. They observed an elevation in extracellular adenosine levels upon activation of astrocytes. However, considering that adenosine concentration itself likely depends on brain states and manipulating both NAc astrocytes and neurons can alter sleep-wake states, it would be better to

simultaneously measure adenosine concentration in both hemispheres of the same animal while manipulating only one hemisphere's neurons or astrocytes to exclude the interference of brain state on adenosine concentration. Additionally, since the title of Fig. 1 is "Astrocytic and neuronal activity in the NAc increases extracellular adenosine levels," Therefore, I believe it would be beneficial if the authors could validate this finding using alternative experimental approaches. For instance, they could replace the chemogenetic methods with optogenetic techniques and substitute the microdialysis method with other adenosine detection methods, such as GRABAdo. Furthermore, I'm curious whether the increase in adenosine concentration induced by activating neurons and astrocytes is dose-dependent. Providing evidence in this regard would further support the authors' claims in the figure's title.

6. The authors seem to have not provided sleep-wake behavior data related to the GFAP-mCherry mice, nor have they presented immunohistochemical results showing the specificity of exogenous gene expression in the GFAP-hM3Dq mice.

7. How do calcium signaling and cAMP levels change in the A2AR-expressing neurons in the NAc upon photolysis of optoA2ARPAM-2 or optoA2ARPAM-1 in vivo? Are the changes consistent with those observed in in vitro conditions?

8. The authors found that intraperitoneal injection of optoA2ARPAM-2 combined with light stimulation in the NAc increased SWS, while stimulation of the ventrolateral preoptic nucleus (VLPO), which also expresses A2A receptors and is known to be involved in adenosine-mediated sleep regulation, did not show the same effect. Can the authors verify the effects of optoA2ARPAM-2 in other brain regions known to be involved in adenosine-mediated sleep regulation through A2A receptors, such as the olfactory bulb?

9. Please provide clear labeling for each panel, such as the missing label for the x-axis in Fig. 3g.

10. Considering that the NAc is involved in motivation-related behaviors, does optoA2ARPAM photolysis in the NAc regulate behaviors other than sleep?

11. How different is the efficiency of A2ARPAM-1 generation during light stimulation between optoA2ARPAM-2 and optoA2ARPAM-1? I understand that these two substances have different abilities to penetrate the blood-brain barrier, resulting in a long-lasting increase in SWS with optoA2ARPAM-2, while optoA2ARPAM-1 has little effect in i.p. conditions. However, why does optoA2ARPAM-1 cause only a minimal increase in SWS under local infusion conditions?

12. The authors concluded that "adenosine may not promote sleep by activating A2ARs on VLPO neurons, as previously proposed" because they did not observe the effect of allosteric A2AR modulation in the VLPO. Does this conclusion appear to lack sufficient evidence?

13. Does the increase in slow-wave sleep (SWS) induced by optoA2ARPAM-2 photolysis in the NAc result from an increase in the number of SWS episodes or an increase in the duration of individual SWS episodes? Is the optoA2ARPAM-2-induced SWS different from the SWS observed under physiological conditions in terms of EEG characteristics?

Reviewer #2 (Remarks to the Author):

This submission utilizes optochemistry to activate the indirect pathway neurons in the nucleus accumbens to induce SWS using a brain-permeable photoactivated positive allosteric modulator of A2ARs.

Major critique:

- The authors states that A2a receptors drives sleep while A1 receptors do not after injection in NAc. This conclusion is based on data collected by injection of a 10-fold higher concentration of the A2a receptor (CGS ,500 μ M) agonist compared with the A1 receptor agonist (CPA, 50 μ M). The high concentration of adenosine that is eliminated multiple ways is understandable, but not the difference in the injection of 2 synthetic agonists. A dose-response curve is needed.
- Injection of 2 μ l bilaterally is a huge volume. The authors must inject a small tracer such as fluorescent to show that the tracer is restricted to NAc after 50 min to argue for their conclusion. What is the injection speed? Were the mice anesthetized?
- Why inject at 19:00?
- The experiments injected bilaterally hM3Dq 109 DREADD-expressing AAV reports on adenosine. Why not on behavior? Also, how is hM3Dq 109 DREADD-expressing changing astrocytic and neuronal activity? Prolonged Gq activation can lead to silences. At least a discussion of prior observations is needed.
- Whole-cell patch-clamp electrophysiology of acutely prepared brain tissue slices. Why not list the region?
- It is no longer ok to base a study on male mice only. Female mice must be included in the key analysis.
- The in vivo analysis of optoA2ARPAM-1 in NAc and its induction of SWS is convincing and carefully performed
- SWS is only a measure of sleep, microarousals, spindles, etc. must be reported and compared between the different manipulations.
- The lack of effect by stimulation of VLPO requires a bit more discussion. Right now the observation is just reported as negative contradicting the literature. What are the possible explanations? Does injection of CGS induce SWA?

Reviewer #3 (Remarks to the Author):

This a very important paper - I have a few reservations, but I am also confident that this team of talented investigators, led by Dr. Lazarus, can easily address them.

First, the group and authors are great, and what they report - mostly technically - is also great, but I'd like to see a bit more physiology. I'll expand on this point

1) For me, the novelty of the paper lies in the use of optochemistry to uncage a positive allosteric modulator of A2ARs locally in the NAc to induce sleep. This is intriguing, but I also think that the allosteric modulator has been previously published, and unless I am missing something, the optochemistry approach appears to be already established. This paper seems to primarily serve as a methodological work (which, no doubt, can move an entire field forward!), as reflected in the abstract.

2) In terms of the mechanism of SWS regulation and the role of astrocytes, the results fall a bit short me. The authors discovered that the activation of glia in the NAc through DREADD activation increases SWS for 6 hours, and this SWS induction by astrocytic activity is dependent on adenosine. While the results are interesting and the effect appears impressive, the authors would be encouraged to conduct inhibitory experiments (necessity) to demonstrate that inhibiting the astrocytes reduces SWS. This would help establish a causal relationship, as it's possible that the SWS induction could result from non-physiological overactivation of astrocytes, unrelated to normal SWS.

3) The authors demonstrated that activation of glia in the NAc via DREADD increases adenosine levels

in the NAc, and similarly, DREADD activation of NAc neurons also has this effect. However, it's unclear why the authors pursued both experiments. The results of neuronal activation diminish the significance of the glia findings, which were the novel data. Additionally, they left some data incomplete. For example, it would be valuable to show the effect of neuronal activation on SWS. It's possible that the authors found no effects on SWS or even a paradoxical waking effect, especially since the NAc contains wake-promoting neurons.

In summary, this paper is methodologically (very, very) interesting, but but I'd like to see something a bit more mechanistic introduced into the narrative. But I'm also worried that I'm being overly critical...it is a super paper and methodology and will be highly cited, no question.

Patrick Fuller

Response to Reviewers

We appreciate the encouraging and constructive comments of the reviewers. We have thoroughly revised the manuscript to address concerns raised by the reviewers. Below are our responses to the reviewers which are highlighted in bold. Moreover, we have indicated all changes with red font in the revised manuscript.

Response to Reviewer 1

The team in their previous work (Korkutata et al., Neuropharmacology, 2019) identified a positive allosteric modulator of A2ARs that can penetrate the blood-brain barrier and named it A2AR PAM-1. They found that both intracerebroventricular and intraperitoneal administration of A2AR PAM-1 increased slow-wave sleep (SWS) in mice, and the sleep-promoting effect of A2AR PAM-1 depended on A2AR. In this manuscript, the authors synthesized two derivatives of A2AR PAM-1, named optoA2ARPAM-1 and optoA2ARPAM-2, which can be "activated" by visible light (~400nm). They validated the effectiveness of these compounds in regulating sleep-wake behavior in nucleus accumbens (NAc) in freely moving animals, providing informative data for the potential application of photopharmacology in basic research and clinical settings. The main findings of this manuscript are as follows:

- 1. Extracellular adenosine signaling in the NAc exerts a sleep-promoting effect, dependent on adenosine A2A receptors rather than A1 receptors.*
- 2. Both astrocytic and neuronal activity in the NAc contribute to the increase in extracellular adenosine levels.*
- 3. Local infusion of optoA2ARPAM-1 in the NAc combined with light stimulation led to an increase in SWS in mice.*
- 4. Intraperitoneal injection of optoA2ARPAM-2 combined with light stimulation in the NAc resulted in increased SWS in mice, and this process depended on A2A receptors.*

Overall, this manuscript is quite interesting, but I have the following questions:

Response: We appreciate the positive comment.

- 1. In Fig.1, I noticed that the adenosine concentration infused in the NAc was 3.5mM. However, if I'm not mistaken, the extracellular adenosine concentration under*

physiological conditions is only in the range of a few hundred nanomolar, and even in some pathological conditions, it can only reach the micromolar level. Therefore, is the increase in SWS induced by adenosine infusion comparable to SWS under physiological conditions (electroencephalogram)? Furthermore, how does the temporal profile of slow-wave sleep (SWS) change in mice after intracerebroventricular infusion of adenosine or adenosine receptor agonists in NAc? Please provide a graph similar to the one shown in Figure 1h.

Response: Due to its rapid clearance by multiple pathways, adenosine was administered at a high concentration, as opposed to agonists or antagonists. In response to the reviewer's suggestion, we added the power spectrum of slow-wave sleep (SWS) after focal NAc injection of adenosine to the manuscript (Supplementary Figure 1c and lines 97-102). Additionally, temporal profiles for focal NAc injections of adenosine, CGS, CPA, and ZM 241385 have been added to the manuscript (Supplementary Figure 1b, e, f).

2. Based on some gain-of-function experiments, the authors concluded the following from Figure 1: 1) Adenosine exerts a sleep-promoting effect in the NAc, dependent on A2A receptors rather than A1 receptors; 2) Both astrocytic and neuronal activity in the NAc contribute to the increase in extracellular adenosine levels. However, these conclusions lack loss-of-function experiments to validate them. For example, what is the effect of adenosine receptor antagonists infused in the NAc on mouse sleep? Which class of adenosine receptors is responsible for blocking the increase in slow-wave sleep (SWS) induced by adenosine infusion? How do inhibiting astrocytic and neuronal activity in the NAc affect extracellular adenosine concentration and mouse sleep?

Response: As suggested by the reviewer, we conducted focal injections of the A2AR antagonist ZM 241385 into the nucleus accumbens (NAc) of wild-type (WT) mice and administered adenosine to A2AR knockout (KO) mice via the same route (Figure 1b and Supplementary Figure 1e-f and lines 104-107). While the antagonist induced wakefulness, adenosine did not elicit an increase in sleep in the A2AR KO

mice. Therefore, we concluded that the sleep-promoting effect of adenosine in the NAc depends on A2ARs.

Additionally, we attenuated astrocyte calcium signaling by expressing human plasma membrane calcium pump isoform 2 and chemogenetically inhibited neurons in the NAc (Supplementary Figure 4 and lines 164-172), which partially influenced extracellular adenosine levels. However, the reduction in astrocytic activity did not lead to changes in SWS sleep. In contrast, we have previously shown that chemogenetic inhibition of A2AR neurons reduces sleep (Oishi Y, et al., Nat. Commun. 2017).

3. Did the authors investigate the role of A1R in the chemogenetic activation-induced sleep increase in NAc astrocytes?

Response: Since the SWS effect of adenosine is attenuated in A2AR KO mice, we refrained from further investigating the involvement of A1R in the NAc.

4. Since NAc adenosine has a sleep-promoting effect, I'm curious about the changes in NAc adenosine concentration during the normal sleep-wake cycle.

Response: Adenosine levels were analyzed under baseline conditions using adenosine GRAB sensors and the obtained results have been incorporated into the manuscript (lines 147-151 and Figure 2b, c).

5. In Fig.1, The authors employed chemogenetic techniques to activate astrocytes in the NAc and simultaneously utilized microdialysis to monitor changes in extracellular adenosine concentrations in the same brain region. They observed an elevation in extracellular adenosine levels upon activation of astrocytes. However, considering that adenosine concentration itself likely depends on brain states and manipulating both NAc astrocytes and neurons can alter sleep-wake states, it would be better to simultaneously measure adenosine concentration in both hemispheres of the same animal while manipulating only one hemisphere's neurons or astrocytes to exclude the

interference of brain state on adenosine concentration. Additionally, since the title of Fig. 1 is "Astrocytic and neuronal activity in the NAc increases extracellular adenosine levels," Therefore, I believe it would be beneficial if the authors could validate this finding using alternative experimental approaches. For instance, they could replace the chemogenetic methods with optogenetic techniques and substitute the microdialysis method with other adenosine detection methods, such as GRABAdo. Furthermore, I'm curious whether the increase in adenosine concentration induced by activating neurons and astrocytes is dose-dependent. Providing evidence in this regard would further support the authors' claims in the figure's title.

Response: We thank the reviewer for his/her comprehensive suggestions. We have incorporated experiments with GRAB sensors for adenosine under baseline conditions and after astrocytic and neuronal manipulations using chemogenetics and an inhibitory Ca pump (lines 141-172 and Figure 2 and Supplementary Figures 2 and 4). While we acknowledge the complexity of adenosine release in the brain and the multitude of experiments that could be performed to explore this issue, we maintain that incorporating optogenetic experiments and the measurement of adenosine levels after dose-dependent activation of astrocytes and neurons is beyond the intended focus of this paper. Nevertheless, we have revised the figure title as follows:

Figure 1 Activation of NAc A2AR by focal injection of adenosine or stimulation of astrocytes induces SWS.

6. The authors seem to have not provided sleep-wake behavior data related to the GFAP-mCherry mice, nor have they presented immunohistochemical results showing the specificity of exogenous gene expression in the GFAP-hM3Dq mice.

Response: We exclusively utilized the GFAP-mCherry mice for the microdialysis experiments. We have previously shown that CNO application does not affect sleep-wake levels in naive mice (Takata Y, et al., J Neurosci, 2018), so we did not

measure sleep in CNO-treated GFAP-mCherry mice to minimize the number of animals used.

The specificity of hM3Dq-mCherry fusion protein expression in astrocytes is shown in Figure 1e, while the extent of transgene expression is shown in Figure 1f. We have revised Figure 1e to show that the mCherry staining as an indicator of the expression of the fusion protein.

7. How do calcium signaling and cAMP levels change in the A2AR-expressing neurons in the NAC upon photolysis of optoA2ARPAM-2 or optoA2ARPAM-1 in vivo? Are the changes consistent with those observed in in vitro conditions?

Response: We sincerely appreciate the reviewer's thoughtful comment. While we find the suggested experiments intriguing, we respectfully contend that they may not provide additional evidence supporting the key conclusions drawn in the paper. In the paper, we used patch-clamp electrophysiology to show that A2ARPAM-1 and optopharmacologically activated A2ARPAM-1 increase the resting membrane potential in A2AR neurons.

8. The authors found that intraperitoneal injection of optoA2ARPAM-2 combined with light stimulation in the NAc increased SWS, while stimulation of the ventrolateral preoptic nucleus (VLPO), which also expresses A2A receptors and is known to be involved in adenosine-mediated sleep regulation, did not show the same effect. Can the authors verify the effects of optoA2ARPAM-2 in other brain regions known to be involved in adenosine-mediated sleep regulation through A2A receptors, such as the olfactory bulb?

Response: Since it is known that A2A receptors in the olfactory bulb regulate REM sleep (Wang Y, et al., *Brain Struct Function*, 2016) and the A2ARPAM-1 only induces SWS (Korkutata M, et al., *Neuropharmacology*, 2019), we felt that the olfactory bulb was not a reasonable target for A2AR optopharmacology. It may be interesting to pursue this idea in a separate study.

9. Please provide clear labeling for each panel, such as the missing label for the x-axis in Fig. 3g.

Response: As suggested by the reviewer, we have added labeling of the x-axis in Figure 4g (previously Figure 3g).

10. Considering that the NAc is involved in motivation-related behaviors, does optoA2ARPAM photolysis in the NAc regulate behaviors other than sleep?

Response: We did not investigate other behaviors after photoactivation of A2ARPAM-1 in the NAc due to its potent suppressive effects, likely attributed to its sleep-inducing properties. To further support this notion, we have included Supplementary Figure 9 and text (lines 277-281) to the manuscript showing that risk-taking behavior in an open field test is suppressed in WT and even in microtubule-associated protein 6 (MAP6) KO mice, a genetic mouse model of schizophrenia/psychosis.

11. How different is the efficiency of A2ARPAM-1 generation during light stimulation between optoA2ARPAM-2 and optoA2ARPAM-1? I understand that these two substances have different abilities to penetrate the blood-brain barrier, resulting in a long-lasting increase in SWS with optoA2ARPAM-2, while optoA2ARPAM-1 has little effect in i.p. conditions. However, why does optoA2ARPAM-1 cause only a minimal increase in SWS under local infusion conditions?

Response: We do not know the efficiency of optoA2ARPAM-1 diffusion through the membrane of the optodialysis probe. The considerable molecular weight of the opto compound poses a potential obstacle to its effective dialysis into the NAc.

12. The authors concluded that "adenosine may not promote sleep by activating A2ARs on VLPO neurons, as previously proposed" because they did not observe the effect of

allosteric A2AR modulation in the VLPO. Does this conclusion appear to lack sufficient evidence?

Response: We agree with the reviewer and have removed the sentence from the discussion.

13. Does the increase in slow-wave sleep (SWS) induced by optoA2ARPAM-2 optolysis in the NAc result from an increase in the number of SWS episodes or an increase in the duration of individual SWS episodes? Is the optoA2ARPAM-2-induced SWS different from the SWS observed under physiological conditions in terms of EEG characteristics?

Response: We have added the SWS and REMS episode numbers and the distribution of SWS episode duration after photoactivation of A2ARPAM-1 in the NAc, a key experiment of this study, to the manuscript (lines 269-276 and Supplementary Figure 8). In terms of EEG characteristics, Supplementary Figure 1c as well as our previous studies (Oishi Y, et al., Nat Commun, 2017, Korkutata M, et al., Neuropharmacology, 2019, Lin Y, et al. Front Pharmacol, 2023) consistently demonstrate that activation of NAc A2AR neurons or administration of A2ARPAM-1 induces sleep that is indistinguishable from physiological sleep.

Response to Reviewer 2

This submission utilizes optochemistry to activate the indirect pathway neurons in the nucleus accumbens to induce SWS using a brain-permeable photoactivated positive allosteric modulator of A2Ars.

Major critique:

- The authors states that A2a receptors drives sleep while A1 receptors do not after injection in NAc. This conclusion is based on data collected by injection of a 10-fold higher concentration of the A2a receptor (CGS ,500 μ M) agonist compared with the A1 receptor agonist (CPA, 50 μ M). The high concentration of adenosine that is eliminated multiple ways is understandable, but not the difference in the injection of 2 synthetic agonists. A dose-response curve is needed.*

Response: As suggested by the reviewer we also administered the A1 receptor agonist CPA at 500 μ M into the NAc. Interestingly, SWS was reduced although the reduction was not statistically significant (lines 102-104 and Supplementary Figure 1d).

- Injection of 2 μ l bilaterally is a huge volume. The authors must inject a small tracer such as fluorescent to show that the tracer is restricted to NAc after 50 min to argue for their conclusion. What is the injection speed? Were the mice anesthetized?*

Response: As suggested by the reviewer, we confirmed the location of the drug infusion in the NAc by injecting the same 2 μ L volume of a 4% solution of fluorescein (lines 91-93 and Figure 1c). We also clarified in the methods section that the solutions were manually injected into unanesthetized animals (lines 589-591):

Microinjections of vehicle or drugs were performed using a tubing-nested Hamilton 10- μ L syringe connected to an internal cannula (62232, RWD Life Science), and the solutions were slowly injected manually into freely behaving mice.

- *Why inject at 19:00?*

Response: We injected at 19:00, when the lights are turned off, to observe sleep induction during the active period of the mice. We have modified the following text in the manuscript (lines 87-92):

We implanted cannulas bilaterally into the NAc of WT or A2AR knockout (KO) mice and analyzed EEG and EMG recordings made after focal NAc injections (2 μ L/side) of vehicle, 3.5 mM adenosine, 500 μ M CGS 21680, and 50 μ M CPA at 19:00 to observe sleep induction when the mice were mostly awake or 6 mM ZM 241385 at 8:30 to observe wake induction when the mice were mostly asleep (Figure 1b).

- *The experiments injected bilaterally hM3Dq DREADD-expressing AAV reports on adenosine. Why not on behavior? Also, how is hM3Dq DREADD-expressing changing astrocytic and neuronal activity? Prolonged Gq activation can lead to silences. At least a discussion of prior observations is needed.*

Response: While we are uncertain about the specific behavior the reviewer is seeking, we suggest the reviewer to refer to our response to question 5 from Reviewer 1 for further clarification.

In response to the reviewer's suggestion regarding the signaling cascades after hM3Dq DREADD activation, as suggested by the reviewer, we add the following text to the discussion of the manuscript (lines 315-320):

Gi- and Gq-GPCR signaling in astrocytes is thought to increase Ca²⁺ activity via IP₃-dependent release of intracellular Ca²⁺^{34–37}, but chemogenetic hM3Dq DREADD activation of cortical astrocytes leads to a long-lasting silent state of Ca²⁺ dynamics after an initial short period of Ca²⁺ activity³⁸. The signaling cascades, possibly Ca²⁺-independent, that lead to sustained adenosine release from NAc astrocytes upon hM3Dq DREADD stimulation, therefore, remains unclear.

- *Whole-cell patch-clamp electrophysiology of acutely prepared brain tissue slices. Why not list the region?*

Response: We have revised the manuscript to clarify that our patch-clamp electrophysiology experiments were conducted on brain slices specifically containing the striatum with the NAc, ensuring a more precise and targeted investigation (lines 204-205).

• It is no longer ok to base a study on male mice only. Female mice must be included in the key analysis.

Response: We highly appreciate the valuable feedback and, in order to minimize the unnecessary use of mice, we have opted not to replicate all experiments in female mice. In accordance with the journal's policy, we explicitly stated in the title and abstract that exclusively male mice were employed for this study, along with a rationale provided in the Reporting Summary.

• The in vivo analysis of optoA2AR/PAM-1 in NAc and its induction of SWS is convincing and carefully performed

Response: We thank the reviewer for this comment.

• SWS is only a measure of sleep, microarousals, spindles, etc. must be reported and compared between the different manipulations.

Response: Microarousal is a distinctive parameter in mice that is not frequently examined in sleep studies. We are curious about the meaningful insights that could arise from observing alterations in microarousal after pharmacologic sleep induction. Consequently, we have chosen not to include it in our study. Instead, as suggested by reviewer 1, we have analyzed the sleep architecture after optopharmacologic stimulation of NAc neurons (lines 269-276 and Supplementary Figure 8). In addition, we already reported that SWS after stimulation of NAc A2AR neurons has a spindle frequency similar to that of control mice (Oishi Y, et

al. Nat Commun, 2017), suggesting that NAc A2AR-expressing neurons do not regulate spindle frequency.

• *The lack of effect by stimulation of VLPO requires a bit more discussion. Right now the observation is just reported as negative contradicting the literature. What are the possible explanations? Does injection of CGS induce SWA?*

Response: After careful consideration of the concern raised by Reviewer 1 in question 12, we agree with both reviewers on this matter. As a result, we have removed the pertinent discussion from the manuscript.

In response to the request from Reviewer 1, we have incorporated the SWS EEG power spectrum following the focal injection of adenosine into the NAc (Supplementary Figure 1c). Our findings lead us to the conclusion that the examination of SWS activity after CGS injection would be redundant.

Response to Reviewer 3

This a very important paper - I have a few reservations, but I am also confident that this team of talented investigators, led by Dr. Lazarus, can easily address them.

First, the group and authors are great, and what they report - mostly technically - is also great, but I'd like to see a bit more physiology. I'll expand on this point

Response: We would like to thank Dr. Fuller for his kind and encouraging comments.

1) For me, the novelty of the paper lies in the use of optochemistry to uncage a positive allosteric modulator of A2ARs locally in the NAc to induce sleep. This is intriguing, but I also think that the allosteric modulator has been previously published, and unless I am missing something, the optochemistry approach appears to be already established. This paper seems to primarily serve as a methodological work (which, no doubt, can move an entire field forward!), as reflected in the abstract.

Response: While the allosteric modulator has been published previously, the in-vivo optochemical approach is novel. One of the major challenges in using optochemistry in the mammalian brain is the potential for phototoxic damage to brain tissue caused by UV light, which is commonly used to activate photolabile protecting groups. In addition, it may be difficult to deliver photocaged compounds across the blood-brain barrier and rapidly uncage them. We are aware of only one study similar to ours published last year in Nature Methods (Ma X, et al, Nature Methods, 2023), which performed in-vivo photopharmacology in the NAc and VTA with a caged mu-opioid receptor agonist that can be uncaged using UV light after direct brain infusion. In contrast, we have developed a photoactivatable allosteric modulator of A2AR that can cross the blood-brain barrier and be rapidly uncaged with visible light. To the best of our knowledge, our study is the first to optochemically induce physiological sleep.

2) *In terms of the mechanism of SWS regulation and the role of astrocytes, the results fall a bit short me. The authors discovered that the activation of glia in the NAc through DREADD activation increases SWS for 6 hours, and this SWS induction by astrocytic activity is dependent on adenosine. While the results are interesting and the effect appears impressive, the authors would be encouraged to conduct inhibitory experiments (necessity) to demonstrate that inhibiting the astrocytes reduces SWS. This would help establish a causal relationship, as it's possible that the SWS induction could result from non-physiological overactivation of astrocytes, unrelated to normal SWS.*

Response: Inhibiting astrocytes is a complex issue. As now discussed in the manuscript (lines 312-320), Gi and Gq GPCR signaling in astrocytes is thought to increase Ca²⁺ activity via IP3-dependent release of intracellular Ca²⁺, so hM4Gi DREADD expression increases activity in astrocytes. Instead, we have attempted to reduce astrocyte calcium signaling by expressing the human plasma membrane calcium pump isoform type 2 (hPMCA2), which constitutively extrudes cytosolic calcium (lines 164-169 and Supplementary Figure 4). While we see some changes in GRABAdo10.0 signaling at the transition from SWS to wakefulness, sleep/wake behavior is not altered (Supplementary Figure 4c-d). It has been shown that hM3Dq DREADD activation of cortical astrocytes leads to a long-lasting silent state of Ca²⁺ dynamics after an initial short period of Ca²⁺ activity (Vaidyanathan, TV, et al. eLife, 2021). Thus, it remains unclear which signaling cascades lead to sustained adenosine release from NAc astrocytes by hM3Dq DREADD stimulation.

Although not specifically recommended by Dr. Fuller, we also performed adenosine measurement after the inhibition of NAc neurons, which resulted in a partial attenuation of GRABAdo signaling (lines 169-172 and Supplementary Figure 4f-h).

3) *The authors demonstrated that activation of glia in the NAc via DREADD increases adenosine levels in the NAc, and similarly, DREADD activation of NAc neurons also*

has this effect. However, it's unclear why the authors pursued both experiments. The results of neuronal activation diminish the significance of the glia findings, which were the novel data. Additionally, they left some data incomplete. For example, it would be valuable to show the effect of neuronal activation on SWS. It's possible that the authors found no effects on SWS or even a paradoxical waking effect, especially since the NAc contains wake-promoting neurons.

Response: The astrocyte-dependent adenosine release has been questioned (Peng, W. et al. Cell Discovery, 2023), so the release of adenosine from astrocytes in the NAc is an important observation. However, we cannot deny that there is also a substantial release of adenosine from neurons, which is likely to contribute significantly to the optoallosteric sleep induction studied in the main part of the manuscript.

As suggested by Dr. Fuller, we performed chemogenetic activation in VGAT-Cre mice to target GABAergic medium spiny NAc neurons (Supplementary Figure 3). Consistent with Dr. Fuller's speculation, activation of GABAergic NAc neurons did not induce sleep, probably due to the opposite sleep/wake effects of direct and indirect pathway NAc neurons. This observation further suggests that astrocytes directly increase adenosine upon activation, rather than stimulating neurons to increase extracellular adenosine, because astrocyte activation induces sleep.

In summary, this paper is methodologically (very, very) interesting, but but I'd like to see something a bit more mechanistic introduced into the narrative But I'm also worried that I'm being overly critical...it is a super paper and methodology and will be highly cited, no question.

Response: We would like to express our sincere gratitude to Dr. Fuller for his positive review of our manuscript and for providing valuable suggestions, along with the other reviewers, to enhance the mechanistic aspects of our work. We are optimistic that the additional experiments performed in this revised manuscript, including novel measurements such as the adenosine GRAB fiber photometry and

additional pharmacologic and chemogenetic experiments will significantly strengthen the depth of our findings.

REVIEWERS' COMMENTS

Reviewer #2 (Remarks to the Author):

The authors have responded well to my comments. New data and discussion have been added and I have no more critique

Reviewer #3 (Remarks to the Author):

This is an outstanding paper that I fully endorse for publication. It will, with certainty, be highly cited.